# A comprehensive resource for *Bordetella* genomic epidemiology and biodiversity studies

Sébastien Bridel [1,7], Valérie Bouchez [1,2,7], Bryan Brancotte [3], Sofia Hauck[4], Nathalie Armatys[1,2], Annie Landier [1,2], Estelle Mühle [5], Sophie Guillot[1,2], Julie Toubiana [1,2,6], Martin C. J. Maiden[4], Keith A. Jolley [4] & Sylvain Brisse [1,2✉]

The genus *Bordetella* includes bacteria that are found in the environment and/or associated with humans and other animals. A few closely related species, including *Bordetella pertussis*, are human pathogens that cause diseases such as whooping cough. Here, we present a large database of *Bordetella* isolates and genomes and develop genotyping systems for the genus and for the *B. pertussis* clade. To generate the database, we merge previously existing databases from Oxford University and Institut Pasteur, import genomes from public repositories, and add 83 newly sequenced *B. bronchiseptica* genomes. The public database currently includes 2582 *Bordetella* isolates and their provenance data, and 2085 genomes (https://bigsdb.pasteur.fr/bordetella/). We use core-genome multilocus sequence typing (cgMLST) to develop genotyping systems for the whole genus and for *B. pertussis*, as well as specific schemes to define antigenic, virulence and macrolide resistance profiles. Phylogenetic analyses allow us to redefine evolutionary relationships among known *Bordetella* species, and to propose potential new species. Our database provides an expandable resource for genotyping of environmental and clinical *Bordetella* isolates, thus facilitating evolutionary and epidemiological research on whooping cough and other *Bordetella* infections.

[1] Institut Pasteur, Université Paris Cité, Biodiversity and Epidemiology of Bacterial Pathogens, Paris, France. [2] National Reference Center for Whooping Cough and other Bordetella Infections, Institut Pasteur, Paris, France. [3] Institut Pasteur, Université Paris Cité,  Bioinformatics and Biostatistics Hub, F-75015 Paris, France. [4] Department of Zoology, University of Oxford, 11a Mansfield Road, Oxford OX1 3SZ, UK. [5] Collection de l´Institut Pasteur, Institut Pasteur, Université Paris Cité, Paris, France. [6] Department of General Pediatrics and Pediatric Infectious Diseases, Université Paris Cité, Hôpital Necker–Enfants Malades, APHP, Paris, France. [7] These authors contributed equally: Sébastien Bridel, Valérie Bouchez. ✉email: sylvain.brisse@pasteur.fr

**B**ordetellae are beta-proteobacteria that are mainly associated with infection in animals and humans, and sometimes retrieved from environmental samples. Of the 16 currently described *Bordetella* species, the medically most important taxa *B. pertussis* (*Bp*) and *B. parapertussis* (*Bpp*), together with *B. bronchiseptica* (*Bbs*), are referred to as the 'classical *bordetellae*' and belong to a single genomic species[1], hereafter named the *B. bronchiseptica* genomic species (BbGS) for convenience. *B. pertussis*, and more rarely *B. parapertussis*, cause whooping cough, characterized by its typical paroxysmal cough, and kill an estimated ~140,000 children annually[2–4]. *Bp* and *Bpp* have evolved from sublineages of *Bbs*[1]. Whereas *Bp* and *Bpp* are human restricted, *B. bronchiseptica* has a broader ecological distribution and causes respiratory infections in a wide range of mammalian hosts including humans. Based on MLST[1] and genomic sequencing[4,5] two distinct *Bbs* groups were named *Bbs* complexes I and IV[1], the last one being more strongly associated to humans.

Other *Bordetella* species were described and can affect humans, although a few were so far only described as environmental. *B. holmesii* can be collected either from blood of septicemic patients or from nasopharyngeal samples of patients with pertussis-like symptoms[6–8]. *B. hinzii* is frequently carried in birds[9,10] and *B. avium* and *B. pseudohinzii* are, respectively, responsible of respiratory disease in poultry or birds[11] and in mice or wild rats[12]. *B. bronchialis*, *B. sputigena* and *B. flabilis* were described from respiratory samples of patients with cystic fibrosis[13,14], and *B. trematum* and *B. ansorpii* were found in infected wounds of immunocompromised patients[4,15,16]. Some *Bordetella* species have been found in environmental samples, such as *B. petrii* or the three recently described species *B. muralis*, *B. tumulicola* and *B. tumbae*[17–19]. In addition to these 16 current *Bordetella* species with standing in the prokaryotic taxonomy, *Bordetella* genogroups were identified from patients with cystic fibrosis based on single-gene *nrdA* sequencing[20]; these are labeled as 'genomosp.' in INSDC bioproject PRJNA385118 and may represent additional *Bordetella* species.

The diversity of lifestyles and medical importance of *Bordetellae* raise important questions about the origins and evolution of pathogenicity in this group[4]. Horizontal gene transfer (HGT) is likely to occur between *Bordetella* species and lineages, as already observed for the O-antigen locus in *B. bronchiseptica*[21], and the environmental species *B. petrii* has genomic islands with atypical G + C content[17] that were associated with the metabolic versatility of this species, whereas in contrast, gene gain is rare or absent in *B. pertussis*, which evolved mainly through gene loss[22]. The use of a unified database of genomes from all *Bordetellae* species would facilitate genomic analyses underlying the phenotypic diversity within this important bacterial genus[23].

Besides the large-scale evolution of *Bordetellae*, the population dynamics within *Bp* are an important topic of epidemiological surveillance, in light of vaccine-driven evolution and the possible emergence and global dissemination of antimicrobial resistance[24–26]. Major evolutionary events in vaccine antigen-related genomic features have been described, including within pertussis toxin subunit A (*ptxA*), the promoter region of the pertussis toxin gene cluster (*ptxP*), and the fimbriae gene *fim3*[24]. In most countries using acellular vaccines, isolates characterized by *ptxP*3 and either *fim3*−1 or *fim3*−2 alleles are currently largely predominant, whereas isolates of the ancestral *ptxP*1 genotype still predominate elsewhere, for example in China[25,27]. In addition, in countries that use acellular vaccines, an increasing proportion of *Bp* isolates are deficient for the production of the immunodominant surface protein pertactin, raising questions on future vaccine effectiveness[28]. The nomenclature of genotypic markers and sublineages needs unification to facilitate collective studies of the global epidemiology and population dynamics in *Bp*.

Harmonization of the genotype nomenclature of bacterial pathogens may be achieved by the gene-by-gene approach called MLST (for multilocus sequence typing), in which allele numbers are uniquely attributed to each locus sequence variant. Bioinformatics platforms that centralize allele definitions and allow curation and open access to genotype nomenclature are critical resources for unified pathogen subtype definitions[29]. Until now, there have been two dedicated *Bordetella* genome sequence databases: Oxford University's PubMLST (unpublished) and Institut Pasteur's BIGSdb[30] platforms. This duality has led to nomenclatural confusion and complexity for users, who may need to consult two distinct databases. With the rapid developments of genomic epidemiology and biodiversity surveys, a common and optimized resource for *Bordetella* evolutionary studies and sequence variants naming is needed. In this work, we set-up such a unified genomic platform at https://bigsdb.pasteur.fr/bordetella/ and expand its features to enable addressing genus-wide evolutionary genomics questions and global tracking of important genotypes within individual species, and particularly within *B. pertussi*s.

## Results

**Genomic resource contents**. After merging Oxford and Pasteur databases, importing publicly available genomes, and adding three genomes of recently described species *B. muralis* (CIP111920[T]), *B. tumulicola* (CIP111922[T]) and *B. tumbae* (CIP111921[T]) that we sequenced, the genomic resource was embodied in the BIGSdb-Pasteur genomic platform for Bordetella (https://bigsdb.pasteur.fr/bordetella/) and comprised representatives all 16 *Bordetella* species currently described in the Prokaryotic taxonomy. The number of genomes varied across species, the most represented ones being the classical *Bordetella* species *B. pertussis* (*n* = 1566), *B. bronchiseptica* (*n* = 626) *and B. parapertussis* (*n* = 197). *Bordetella holmesii* (*n* = 59) and the other species were represented by fewer genomes, including some by only one genome, such as *B. flabilis* or *B. sputigena*. The 2085 public genomes are listed in BIGSdb project "Public Genomes" (project i.d. 27).

**Phylogenetic analysis of the genus**. The average nucleotide identity (ANI) metric is now universally used as a guide to define novel bacterial species when genomic sequences are available[31]. ANI values between representative genomes of the *Bordetella* genus (Supplementary Data 1; Supplementary Fig. 1) showed that most currently described species differ by >5% nucleotide positions. A well-known exception is represented by the classical *Bordetella* members, which present ANI values between 98.3% and 99.4% among them, and which belong to a single genomic species, which we have labeled here the *B. bronchiseptica* genomic species (BbGS).

We defined genomic species as groups of genomes with >95% ANI and numbered these from 1 to 26, by matching these numbers with a previous single gene-based genogroup nomenclature[20]; 6 numbers were not attributed to avoid confusion with this previous nomenclature (see Table 1 and Supplementary Data 2). We note that previously labelled genogroup 8 isolates belong to two different genomic species (genomic species 8 and 18), whereas genogroup 6 corresponds to lineage II of the *B. bronchiseptica* genomic species (see below). Several previously attributed species names would need reevaluation; for example, *B. petrii* type strain and '*B. petrii*' isolate J51 had only 85.14% ANI, indicating that the latter should be considered as belonging to a distinct species, which remains to be

**Table 1 Bordetella genomic species defined in this work.**

| Genomic species[a] | Former denomination | No. of isolates | No. of genomes | Reference strain (BIGSdb ID) | Accession number(s) | Genome status | Original description |
|---|---|---|---|---|---|---|---|
| Bordetella genomic species 1 | Bordetella genogroup 1 | 6 | 2 | AU9795 (266) | ASM226143v1; GCF_002261435.1; SAMN06767132 | 7 contigs | Ref. [20] |
| Bordetella genomic species 2 | Bordetella genogroup 2 | 8 | 2 | AU8256 (271) | ASM226134v1; GCF_002261345.1; SAMN06767130 | 7 contigs | Ref. [20] |
| Bordetella genomic species 4 | Bordetella genogroup 4 | 3 | 2 | AU14378 (287) | ASM226129v1; GCF_002261295.1; SAMN06767136 | 19 contigs | Ref. [20] |
| Bordetella genomic species 5 | Bordetella genogroup 5 | 2 | 2 | AU14646 (290) | ASM226147v1; GCF_002261475.1; SAMN06767137 | 13 contigs | Ref. [20] |
| Bordetella genomic species 7 | Bordetella genogroup 7 | 4 | 3 | AU18089 (299) | ASM226126v1; GCF_002261265.1; SAMN06767141 | 8 contigs | Ref. [20] |
| Bordetella genomic species 8 | Bordetella genogroup 8 | 1 | 1 | AU19157 (302) | ASM211968v1; GCF_002119685.1; SAMN06767142 | Complete | Ref. [20] |
| Bordetella genomic species 9 | Bordetella genogroup 9 | 2 | 2 | AU14267 (304) | ASM211974v1; GCF_002119745.1; SAMN06767135 | Complete | Ref. [20] |
| Bordetella genomic species 10 | Bordetella genogroup 10 | 2 | 1 | AU16122 (306) | ASM226122v1; GCF_002261225.1; SAMN06767138 | 5 contigs | Ref. [20] |
| Bordetella genomic species 11 | Bordetella genogroup 11 | 1 | 1 | AU8856 (308) | ASM226121v1; GCF_002261215.1; SAMN06767131 | 4 contigs | Ref. [20] |
| Bordetella genomic species 12 | Bordetella genogroup 12 | 1 | 1 | AU6712 (309) | ASM226135v1; GCF_002261355.1; SAMN06767128 | 3 contigs | Ref. [20] |
| Bordetella genomic species 13 | Bordetella genogroup 13 | 1 | 1 | AU7206 (310) | ASM211966v1; GCF_002119665.1; SAMN06767129 | Complete | Ref. [20] |
| Bordetella genomic species 18 | Bordetella genogroup 8 | 1 | 1 | AU21707 (303) | ASM226142v1; GCF_002261425.1; SAMN06767143 | 3 contigs | Ref. [20] |
| Bordetella genomic species 19 | Bordetella ansorpii | 1 | 1 | H050680373 (2171) | FKIF01; GCA_900078705.1; SAMEA3906486 | 10 contigs | This study |
| Bordetella genomic species 20 | Bordetella sp. | 1 | 1 | FB-8 (1021) | ARNH01; ASM38218v1; DSM24873; GCA_000382185.1; GCF_000382185.1; PRJNA187096; PRJNA200428; SAMN02440498 | 6 contigs | This study |

**Table 1 (continued)**

| Genomic species[a] | Former denomination | No. of isolates | No. of genomes | Reference strain (BIGSdb ID) | Accession number(s) | Genome status | Original description |
|---|---|---|---|---|---|---|---|
| Bordetella genomic species 21 | Bordetella sp. | 1 | 1 | H567 (2694) | ASM170429v1; GCF_001704295.1; SAMN03940794 | Complete | This study |
| Bordetella genomic species 22 | Bordetella sp. | 1 | 1 | N (2693) | ASM143339v1; GCF_001433395.1; SAMN04219331 | Complete | This study |
| Bordetella genomic species 23 | Bordetella petrii | 1 | 1 | BT-1-9.2 (2697) | ASM1774559v1; GCF_017745595.1; SAMN13166181 | 68 contigs | This study |
| Bordetella genomic species 24 | Bordetella sp. | 1 | 1 | DE0060 (2695) | ASM767996v1; GCF_007679965.1; SAMN11792220 | 24 contigs | This study |
| Bordetella genomic species 25 | Bordetella petrii | 1 | 1 | BOR01 (2696) | ASM1921888v1; GCF_019218885.1; SAMN19994144 | 29 contigs | This study |
| Bordetella genomic species 26 | Bordetella petrii | 1 | 1 | BMC_SL_3 (2698) | ASM1735624v1; GCF_017356245.1; SAMN18228559 | 14 contigs | This study |
| BbGS lineage II[b] | Bordetella genogroup 6 | 9 (20) | 2 (13) | AU22978 (297) | ASM21970v1; GCF_002119705.1; SAMN06767144 | Complete | This study |
| n/a[c] | Bordetella genogroup 16 | 1 | 0 | TRE152202 (402) | n/a | n/a | n/a |
| n/a[c] | Bordetella genogroup 17 | 1 | 0 | AU30427 (383) | n/a | n/a | n/a |

[a]Genomic species numbers were inherited from genogroup numbering (see ref. [20]) where possible. Genogroups 3, 14 and 15 belong to the recently defined *B. bronchialis*, *B. sputigena* and *B. flabilis*; therefore, the corresponding numbers were not used for genomic species.
[b]Represents a separate *B. bronchiseptica* clade that may be considered as a genomic species per se.
[c]Numbers 16 and 17 were not used for genomic species, as genogroups 16 and 17 were defined previously based on single gene sequencing and might correspond to new genomic species, but no genomic sequence is available yet.

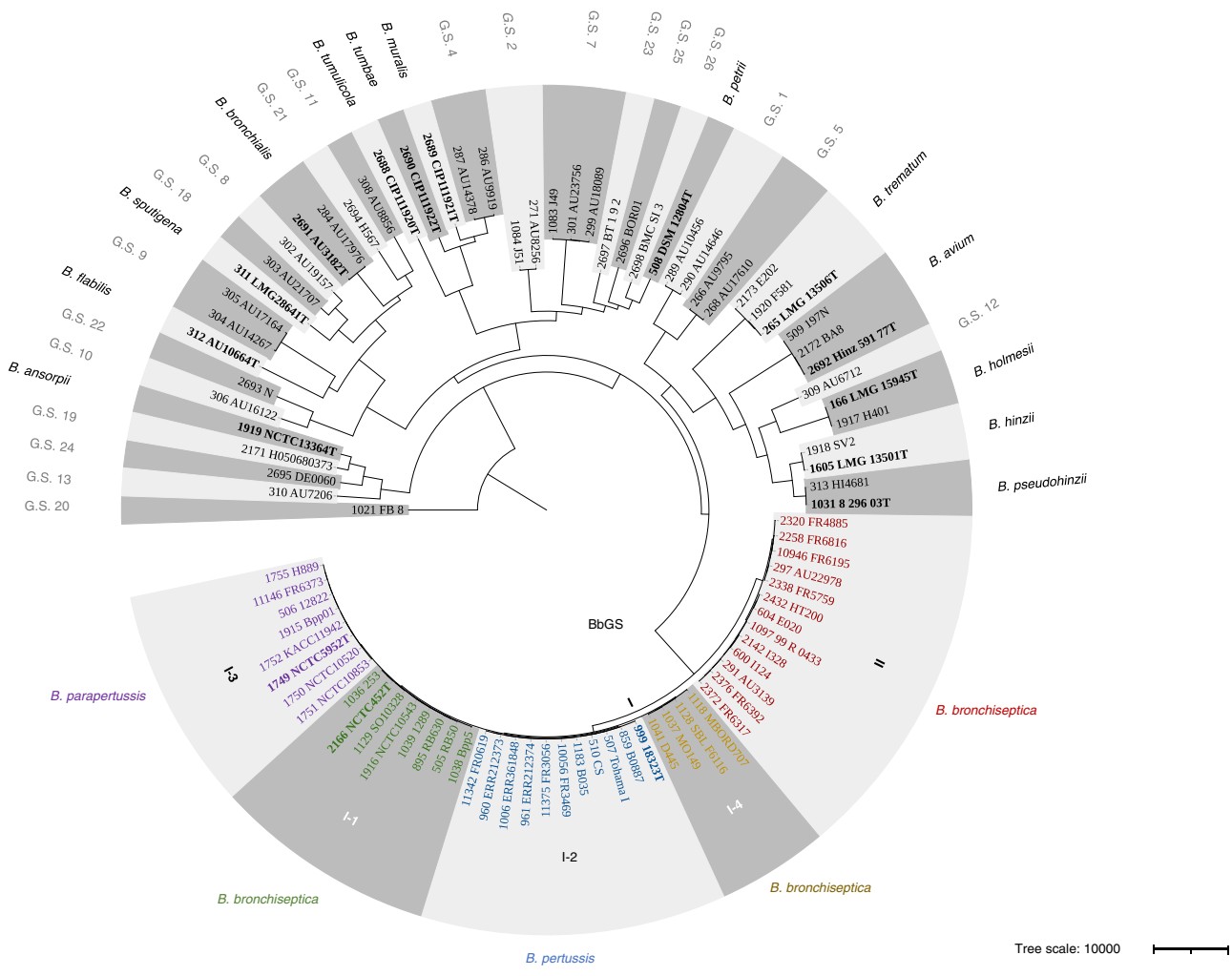

**Fig. 1 Phylogenetic analysis of the *Bordetella* genus.** The phylogenetic tree was obtained based on the concatenated multiple sequence alignments of the 1415 core gene sequences from the *cgMLST_genus* scheme; recombination was accounted for using Gubbins. The tree was rooted on the branch leading to isolate FB-8, which was the earliest branching genome in a phylogenetic analysis that included the external group *Ralstonia solanacearum* isolate IBSBF1503 (Supplementary Fig. 2). Leaves are labelled with the identifier of the isolate in the BIGSdb database, followed by the isolate name. Genomic species are labelled as G.S. An interactive iTOL version of the tree can be accessed at: https://itol.embl.de/shared/1l7Fw0AvKOoCF. Source data are provided as a Source Data file.

defined. Likewise, '*B. ansorpii*' isolate H050680373 had an ANI of 93.48% with *B. ansorpii* type strain NCTC13364[T] (=SMC-8986[T]).

The genus-wide cgMLST scheme loci covered a ~20–30% fraction of the entire length of *Bordetella* genomes, depending on species (Supplementary Data 3). A phylogenetic analysis of representatives of the deep branches of the *Bordetella* genus diversity (see the "Methods" section) was performed based on the alignments of the cgMLST gene loci sequences (Fig. 1). The dataset included representatives of the 16 currently described *Bordetella* species, and the 20 additional *Bordetella* genomic species defined above. Higher-level clades comprising several (genomic) species could be recognized. As expected, the three classical *Bordetella* species were grouped in a common clade. A second clade comprised *B. holmesii*, *B. avium*, *B. hinzii* and *B. trematum*, consistent with Linz et al.[4], and also *B. pseudohinzii* and *Bordetella* genomic species 1, 5, and 12[20]. A third major clade comprised *B. petrii* and genomic species 2, 7, 23, 25, and 26. The three novel species *B. muralis*, *B. tumbae* and *B. tumulicola* and genomic species 4 were part of a single clade, which was distantly associated to the *B. petrii* clade, consistent with previous 16S based phylogenetic analyses[19]. *B. flabilis, B. sputigena* and *B. bronchialis* were found in a fourth, more distant clade together

with genomic species 8, 9, 10, 11, 18, 21, and 22. Finally, the earliest branching *Bordetella* species was *B. ansorpii*, as previously observed[4], and it was associated with genomic species 13, 20, 19, and 24.

**Phylogenetic diversity within the *Bordetella bronchiseptica* genomic species.** We gathered all unique isolates identified as *B. bronchiseptica* and added 83 *B. bronchiseptica* isolates from France that were sequenced for this study into a single public project (project i.d. 24: *B. bronchiseptica* phylogeny, see Supplementary Data 4). After automated filtering of genomes exhibiting more than 25% of missing data, a phylogenetic analysis was performed with representatives of *B. pertussis* and *B. parapertussis* and revealed a primary separation into two deep lineages (Fig. 2 and Supplementary Fig. 3). We define the minor one as lineage II of *B. bronchiseptica*. Lineage II comprised 13 isolates, including HT200, collected from a natural spring water in India and previously recognized as being atypically distant from other *Bbs* isolates[32]. *Bbs* lineage II also comprised isolates from the USA collected from patients with cystic fibrosis and previously described as genogroup 6[20], and isolate I328 previously identified

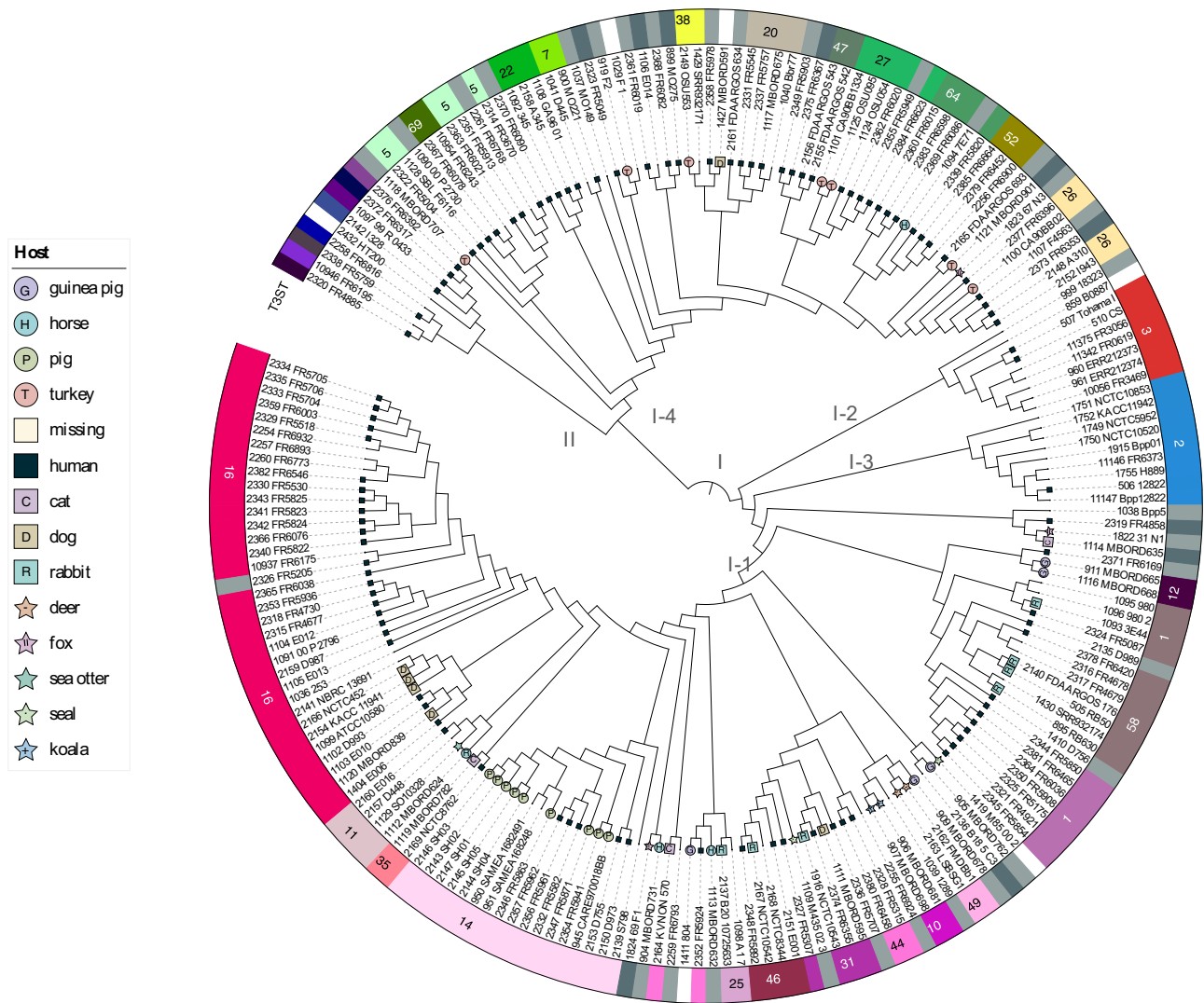

**Fig. 2 Cladogram of the *Bordetella bronchiseptica* genomic species.** The analysis was performed with 186 *B. bronchiseptica* genomes, and representatives of the phylogenetic diversity of *B. pertussis* (9 genomes) and *B. parapertussis* (9 genomes). The recombination-purged concatenated multiple sequence alignment of 1415 core gene loci (*cgMLST_genus* scheme) was used. The tree is rooted on lineage II, which is the most divergent clade. Branch lengths were not used to ease readability of groups and isolates; see Supplementary Fig. 5 for the corresponding phylogram. For each isolate, the host is represented using a leaf symbol, where available (see key in Supplementary Fig. 5). The numbers corresponding to T3SS sequence types are indicated along the external circle around the tree; only the identifiers of main T3STs are indicated. An interactive iTOL version of the tree can be accessed at: https://itol. embl.de/shared/1l7Fw0AvKOoCF. Source data are provided as a Source Data file.

as an atypical *B. bronchiseptica*[33]. Six isolates from France, collected from respiratory samples from adults (mean age: 71.7 years), also belonged to *Bbs* lineage II.

In turn, *Bbs* lineage I comprised four sublineages, which we define as sublineages I-1–I-4 in order to mirror the previous 7-gene MLST-defined clonal complexes I (here, sublineage I-1), II (Bp, sublineage I-2), III (Bpp, sublineage I-3), and IV (sublineage I-4). The ANI values between *Bbs* lineage II and the four sublineages of lineage I ranged from 95.09% to 95.72%, indicating that *Bbs* lineage II may be considered as a species distinct from lineage I, which itself equates to the BbGS or 'classical *bordetellae*'.

A separate phylogenetic analysis performed using the cgMLST_pertussis scheme, instead of the genus-wide scheme, provided a nearly identical phylogenetic tree of *Bbs* isolates (data not shown). However, we recommend using the cgMLST_genus scheme for *Bbs* phylogenetic analyses, as a lower number of uncalled alleles was observed than for the cgMLST_pertussis scheme (Supplementary Fig. 4).

**Genotyping of vaccine antigen and virulence-associated genes of *Bordetella bronchiseptica*.** The allele diversity and presence/absence of vaccine antigens or virulence-associated loci were investigated within *Bbs*. For convenience, the loci were grouped into schemes labeled as Bp-vaccine antigens, Autotransporters, Phase genes, Other Toxins, and T3SS.

Type 3 secretion system (T3SS) genes are known to be present within the classical *Bordetella* species[5]. However, expression of these genes differ in vitro[34]: although *Bbs* and ovine *Bpp* produce T3SS effectors, such as BteA, T3SS expression is blocked in human adapted *Bp* and *Bpp* at post-transcriptional level[3,34]. The insertion of an alanine in position 503 of BteA in *Bp* leads to the attenuation of cytotoxicity[35]. This insertion was found in 8 of 11 isolates of *Bbs* sublineage II-2 (*Bp*), corresponding to alleles 1 and 6.

We devised a nomenclature of sequence types based on the combination of alleles at the loci *bteA*, *bopB*, *bopD*, *bopN*, and *bsp22*; these are referred to as T3SS sequence types (T3ST). For

genes *bteA*, *bobP*, *bopD*, *bopN* and *bsp22*, most alleles were specific of (sub)lineages with some minor exceptions. *Bbs* sublineages are characterized by diverse but unique sets of T3STs (Figs. 2 and S5): sublineage I-1 comprised 31 T3STs (the main ones being T3ST16 and T3ST14), whereas sublineage I-4 comprised 33 other T3STs. Interestingly, sublineage I-2 (*Bp*) isolates all belonged to T3ST3. Isolates of lineage II each had a distinct T3ST. Hypervirulent *Bbs* isolates have been defined previously based on an enhanced activity of the T3SS, for example isolate 1289 (BIGSdb ID: 1039; T3ST: 19) in sublineage I-1 or isolate Bbr77 (BIGSdb ID: 1040; T3ST: 20) in sublineage I-4[36,37]; however, they do not share the same T3ST. The T3ST nomenclature therefore provides a classification system useful for distinction of *B. bronchiseptica* isolates, particularly within the broad (sub)lineages I-1, I-4, and II.

The diversity observed at loci of the other virulence or vaccine-antigen schemes were largely congruent with T3SS variation, with alleles being unique to *Bbs* subgroups within sublineages I-1 and I-4 (Supplementary Note 1 and Supplementary Fig. 5). We also noted a high diversity of alleles within *B. bronchiseptica* lineage II, and that *fim2, ptxP, dnt, tcfA, vag8, prn* or *bipA* had no allele tagged according to our stringent criteria. When relaxing these criteria, we observed that *ptxP, prn* and *dnt* were absent in *Bbs* lineage II, whereas *fim2, bipA* and *tcfA* presented partial matches (<90%) with existing alleles in almost all isolates.

**Allele diversity in *Bordetella pertussis*.** A phylogeny of 124 *Bp* isolates (selected to be representative of main *ptxP* branches and with a focus on macrolide resistance, see the "Methods" section, "Phylogenetic analysis", for further details) was built using the 2,038 loci of cgMLST_pertussis scheme (Fig. 3). This tree enables the visualization of important evolutionary landmarks in genes associated with vaccine antigens. The early-diverged *ptxP2* branch, which corresponds to *Bp* lineage II-a as defined by Bart et al.[24], was characterized by alleles *ptxA4* and *fim2-2*. The remaining isolates corresponded to lineage IIb[24]. One of the earliest changes within this lineage was the *ptxA1* mutation (whereas most vaccine strains express *ptxA2*); the *ptxA1* branch was itself subdivided into *ptxP1* and *ptxP3* branches[24]. In our dataset, considering isolates collected after 2008, *ptxP1* isolates mainly originated from China, whereas *ptxP3* isolates were predominantly from France and the USA (Supplementary Fig. 6). The *ptxP3* branch is itself subdivided by allelic variation of the *fim3* gene (*fim3-1* and *fim3-2* genotypes). Variation observed at the ptxA1, *ptxP* and *fim3* loci has previously been attributed to vaccine-driven evolution[24].

Pertactin (PRN) is another important vaccine antigen of *B. pertussis*. Even though this adhesin is not expressed in a large fraction of currently circulating isolates in countries that use acellular vaccines[28,38,39], its genetic variation also marks important phylogenetic subdivisions[24]. We included the *prn* locus in our database within the Autotransporters scheme (scheme i.d.: 12). Most isolates (*n* = 30) present either *prn1* or *prn2* alleles (Fig. 3). Based on French isolates with phenotypic expression data (Western blot; *n* = 44), most (8 out of 9) *prn*-negative isolates were genotyped with allele 0, corresponding to a missing or disrupted allele.

We also defined a *Bp* vaccine antigens scheme, which groups the loci *ptxP, ptxABCDE, fim2, fim3* and *fhaB2400_5550*. *Bp* vaccine antigens sequence types (BPagST) were defined for this scheme, corresponding to the unique combinations of alleles at the individual loci. These STs were distributed along the *Bp* tree (Fig. 3). For example, BPagST34 marked the sublineage with allele *fim3−4*, whereas BPagST9 corresponded to most isolates of the *fim3−2* branch.

Locus sequence variation within other schemes (Other toxins, Autotransporters, T3SS and Phase biology schemes) are detailed in the Supplementary Note 2.

**Macrolide resistance markers in *Bordetella pertussis*.** To detect alleles associated with macrolide resistance, *Bp* isolates were screened with the dedicated macrolide resistance scheme, which comprises the gene coding for 23S rRNA, and *fhaB*. Allele 13 of the 23S_rRNA locus, characterized by the A2047G mutation, is specific for macrolide-resistant isolates, whereas alleles 1, 2, and 20 correspond to susceptible ones. Allele 13 was found in 46 of 51 isolates from China[25] and in two isolates previously described as erythromycin resistant: one from USA (ATCC:BAA-1335D-5[40]) and one from France (FR4991[41]). When checking our entire database, which contained 1566 *Bp* isolates, no additional resistant isolate was found.

In a previous study, Xu et al.[25] showed that allele *fhaB3* characterized by the mutation C5330T was associated with macrolide-resistant isolates from China. Here, macrolide-resistant isolates from China shared the same allele for *fhaB* (full): fhaB-39. Only one exception was apparent, for isolate L14404 which had a 15-nucleotide insertion at position 6959, defined as allele *fhaB−40*. We also defined loci for internal regions of gene *fhaB* and followed the nomenclature from literature[25]: alleles observed in isolates of the *fhaB3* branch had fhaBy-3190_7183 allele 3 and fhaB-2400_5550 allele 3. However, the correspondence between 23S rRNA and fhaB was not complete: whereas all isolates from China with 23S rRNA allele 13 have the *fhaB3* allele (Supplementary Fig. 6), the opposite was not true, as some isolates (PT2019, ERR030030, ERR361878, SRR1610553) have the fhaB-2400_5550 allele 3, but 23S rRNA allele 1. Hence, the *fhaB3* allele marks a broader lineage, of which a subset of isolates possesses the macrolide-resistance 23S rRNA allele 13.

As observed in the phylogenetic tree (Fig. 3), macrolide-resistant *Bp* isolates from China all fell in the branch characterized by fhaB2400_5500 allele 3 and 23S rRNA allele 13). The two other resistant isolates, collected in the USA and France, belonged to the *ptx*P3/*fim*3-2 branch and were not grouped together, indicating three independent evolutionary origins of macrolide resistance. The macrolide resistant isolates from the USA and France both displayed fhaB2400_5500 allele 1.

Using the vaccine antigens scheme, we note that BPagST37 (allele 3 for *fhaB2400_5550* locus) corresponded fully to the macrolide resistant *fhaB3* lineage of Xu et al.[25]. The macrolide-resistant isolate from France was characterized by BPagST68, whereas no BPagST could be determined for the macrolide-resistant isolate from the USA, because its allele at the *fim3* locus was not defined due to a frameshift. These observations suggest that the BPagST sequence types will be useful for tracking the spread of specific sublineages of *Bp*, including macrolide-resistant ones.

**BIGSdb data visualization functionality.** BIGSdb offers several possibilities to visualize data from the genomic database[29] including the use of built-in plugins that connect to external tools such as GrapeTree[42] or iTOL[43]. We illustrate the use of GrapeTree by visualizing the diversity of cgMLST profiles of *B. pertussis, B. parapertussis* and *B. bronchiseptica* genomes (Supplementary Fig. 7). This analysis illustrates that the main lineages of *B. bronchiseptica* are distinguished using cgMLST profile-based comparisons, and how the diversity of each sublineage is internally structured into cgMLST types. In turn, the iTOL plugin can be used to build phylogenetic trees based on sequence alignments; we illustrate this with the *nrdA* gene, which was initially used to define

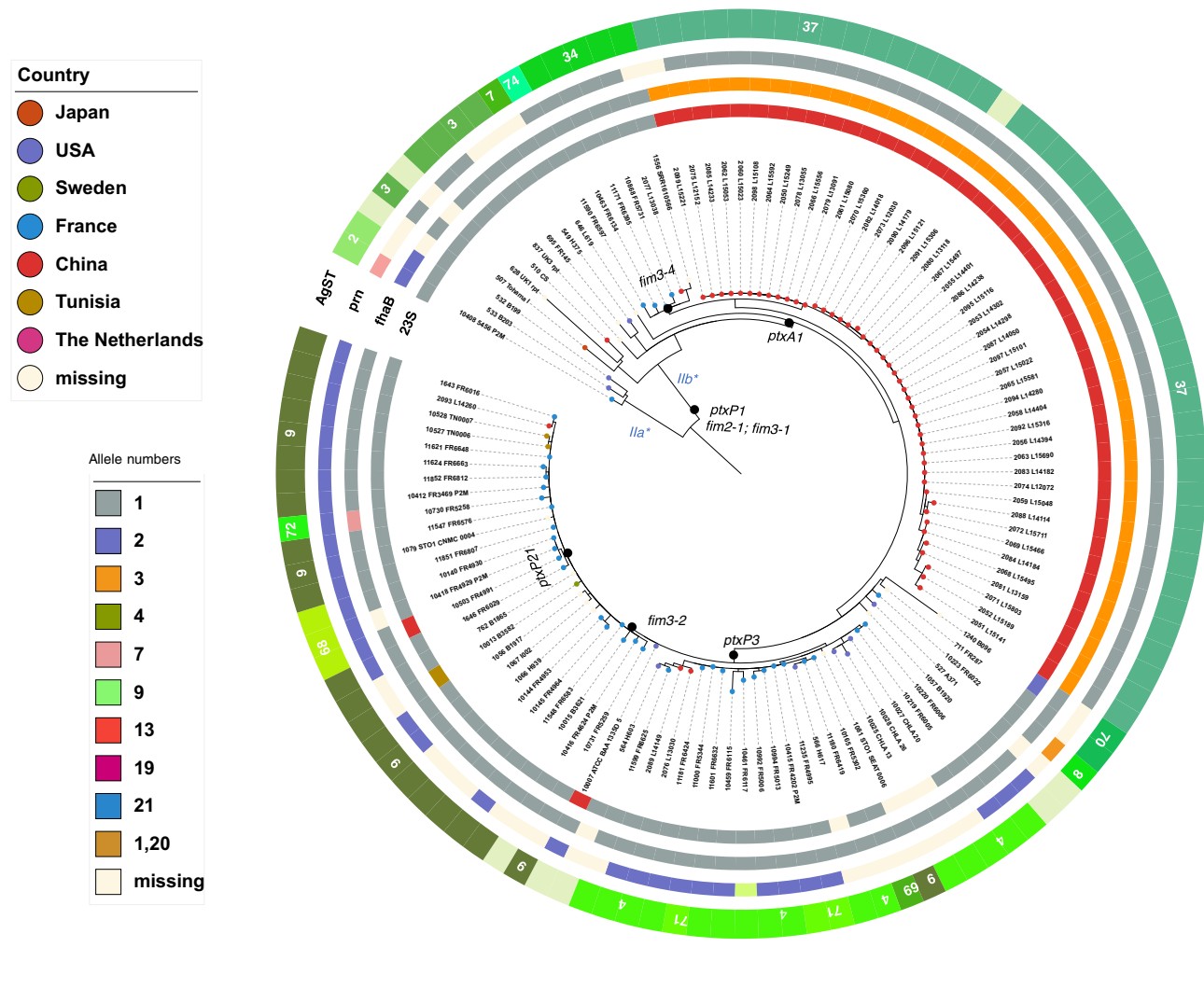

**Fig. 3 Phylogenetic analysis of _Bordetella pertussis_.** The phylogenetic tree was obtained based on the recombination-purged concatenated multiple sequence alignments of the 2038 core genome loci of the _cgMLST_pertussis_ scheme. The distribution of macrolide resistance is indicated. The three outer circles represent (from innermost): _23S_rRNA_ alleles, _fhaB_ alleles, _prn_ alleles and the vaccine antigen sequence types (AgST). Lineages IIa and IIb were defined in Bart et al. 2014 and are labelled in blue. An interactive iTOL version of the tree can be accessed at: https://itol.embl.de/shared/1l7Fw0AvKOoCF. Source data are provided as a Source Data file.

genogroups[20]. The _nrdA_-based phylogenetic tree (Supplementary Fig. 8) shows how this gene distinguishes the novel genomic species that were defined in this work and those that correspond to previously defined genogroups.

## Discussion

We have set-up a unified genomic platform, available at https://bigsdb.pasteur.fr/bordetella/, by merging the data from the two previously existing PubMLST and Pasteur BIGSdb-_Bordetella_ databases. This merger has enabled allelic nomenclature unification and has led to reassemble on the same platform, genotyping schemes that will facilitate the study of _Bordetella_ diversity at different phylogenetic breadths. We have also expanded the genomic resource with public sequences and sequenced genomes from the French NRC. The resulting unified platform for _Bordetella_ genotyping will ease the future curation and expansion of _Bordetella_ genomic data by a community of users and curators.

Two complementary cgMLST schemes are available on the genomic platform. While the _cgMLST_genus_ scheme should be preferred for genus-wide phylogenetic analyses and within individual species including _B. bronchiseptica_, the _cgMLST_pertussis_ scheme comprises more loci that are conserved within _Bp_, and its added discriminatory power will be useful for strain identification and comparison within _Bp_. Recently, a whole-genome MLST (wgMLST) scheme comprising even more loci ($n = 3506$; 1822 of which are common with our _cgMLST_pertussis_ scheme) was proposed for Bp subtyping, and concordant results were obtained for classification of outbreak and sporadic isolates from the USA using the cgMLST and wgMLST schemes[44]. As the later provides slightly higher discrimination among Bp isolates than the _cgMLST_pertussis_ scheme, it might provide valuable additional genotyping information for local transmission investigations and may be incorporated into the unified platform in the future.

We provided examples of applications of the genomic database and its associated genotyping schemes, from the analysis of the genus-level diversity of species, to isolate genotyping within _Bp_. Regarding the phylogenetic structure of _Bordetella_, we provide a complete picture of the relationships among the 16 _Bordetella_ species that have current standing in the Prokaryotic taxonomy

and confirm the existence of 20 additional putative species. This work shows that the *Bordetella* taxonomy will need future updates, and the unified *Bordetella* genomic resource should facilitate the recognition of isolates belonging to the same putative novel *Bordetella* taxa and the definition of their microbiological, ecological, or pathogenic properties.

An important finding of this work is the identification of a novel *B. bronchiseptica* lineage, which we provisionally named *Bbs* lineage II. The divergence of some *B. bronchiseptica* isolates was previously recognized by Spilker et al. (2014)[20], Weigand et al. (2019)[33] and Badhai et al. (2020)[32]. This lineage is clearly distinct from the classical *Bordetella*, which we therefore propose to define strictly as the BbGS, i.e., encompassing the previously described *Bbs* lineages I-1 and I-4 together with *Bp* and *Bpp*. The average ANI between *Bbs* lineage II and the BbGS members lies within the species definition threshold range (95–96%), and the relevance of elevating *Bbs* lineage II as a taxonomic species or subspecies separate from *B. bronchiseptica* remains to be defined. Virulence factors content and vaccine antigens sequence variants were highly variable and unique to lineage II, and all but one isolate from this lineage were collected from humans. Defining the pathogenic potential and epidemiology of *Bbs* lineage II is an important topic of future research.

A rarely isolated subgroup of *Bpp* causes pneumonia in sheep and has been referred to as $Bpp_{ov}$ in order to mark its distinctness from $Bpp_{hu}$, the human *Bpp*[45]. $Bpp_{ov}$ remains an under-studied lineage within the BbGS. So far, very few isolates of ovine *Bpp* have been reported, mainly from New Zealand and Scotland, and all were collected before 2006[45–47], suggesting that this lineage is very rare, if at all still circulating. Bpp5 is the only $Bpp_{ov}$ representative for which a genome sequence is currently available. Even though it represents a unique genetic variant within the BbGS (Figs. S3 and S7), more genomes of $Bpp_{ov}$ would be needed to define its phylogenetic origins and unique genomic features.

*Bp* is the medically most important *Bordetella* species, being the most frequent cause of important infections in humans. The unified *Bordetella* genomic platform provides specific tools for characterizing genes linked to *Bp* vaccine antigens, virulence, and antimicrobial resistance. These were organized into several genotyping schemes that might be used separately for specific biological or epidemiological questions. Where available, we defined allele identifiers in accordance with their literature numbering. This will disambiguate especially the genotyping of vaccine antigens within *Bp*. The population of *Bp* has been noted to evolve in response to the selective pressure exerted by vaccination, in particular in countries using the acellular vaccines[24,48]. The proposed vaccine schemes will help keeping track of the circulating isolates and defining the prevalence of particular sublineages, including those that are deficient for pertactin production[28,38,39].

We also created a genotyping scheme dedicated to macrolide resistance, the possible dissemination of which is an important point of public health concern. To enable easy identification of the macrolide-resistant sublineage that is currently highly prevalent in China, we defined a nomenclature of alleles of the associated marker *fhaB*, providing a convenient way to distinguish 'out of China' dissemination of the *fhaB3* sublineage, from parallel evolution of macrolide resistance in other locations.

The following limitations of the present work must be considered. First, there is an imbalance towards *B. pertussis* genomic sequences from countries that use acellular vaccines. The genomic database currently contains only few genomes of isolates from Africa (0.2%), Asia (1.9%) or Oceania (2.1%) whereas a majority comes from Europe (58.7%). Still, historical isolates from Europe and the USA, and more recent isolates from China[24,25,49], are available and represent a diversity of sequences from the *ptxP1*

branches that are typically prevalent in countries using whole-cell vaccines. In the future, efforts should be made to generate and incorporate more genomic sequences from world regions that are currently underrepresented, and the genomic database and attached analytical tools may promote their integrated analysis. A second limitation is that several genotyping schemes are not applicable to the entire breadth of phylogenetic diversity of the database; for example, the T3SS scheme is meant only for genotyping BbGS isolates. More dedicated genotyping schemes can be incorporated in the future to address specific needs, including for population biology studies of the so far little sampled environmental species. Third, as BIGSdb is a gene-by-gene analytic platform that relies on genomic assemblies, it does not enable the investigation of more complex variation such as heterozygosity among multiple gene copies, large gene disruptions, nor does it allow accurate recombination detection, particularly in intergenic regions and accessory genes, which are not included in the cgMLST schemes. This has implications specifically for the 23S rRNA and pertactin genes mutation studies. Erythromycin-resistant strains of *B. pertussis* can display heterozygous *rrn* operons, with only one or two copies having the A2047G mutation[26,40]. As assembled sequences represent majority consensus, we infer that 23S rRNA allele 13 is observed when at least two of the three *rrn* copies are mutated. As mutation in a single *rrn* copy is already sufficient to raise the MIC to >64 µg/mL, such macrolide-resistant strains may be missed using consensus assemblies. Regarding pertactin deficiency, inactivation of pertactin can be caused by insertion sequences integration and large genomic inversions, which BIGSdb is not designed to characterize; however, these events are tagged as allele 0 and can be investigated with external tools.

In conclusion, the unified *Bordetella* genomic resource is intended to facilitate and harmonize the genotyping of isolates of this important bacterial genus and should represent a useful tool particularly for genomic epidemiology consortia unfamiliar with genomic analyses. It provides a uniformization of genetic variants designations, which will clarify the communication on genotypes and will enable a collective understanding on the biodiversity and epidemiology of *Bordetella*. We provide a timely solution for genomic studies of *Bordetella* pathogens, and notably *Bp*, the reemergence of which is partly caused by evolutionary and epidemiological dynamics of public health concern, including partial vaccine escape and antibiotic resistance emergence.

## Methods

**DNA preparation and genomic sequencing.** Genomic sequencing was performed for 83 French *B. bronchiseptica* isolates, all except 5 were of human origin. These 83 isolates were collected between 2007 and 2020. We also sequenced the type strains of *B. tumulicola*, *B. muralis* and *B. tumbae*.

Isolates were grown at 36 °C for 72 h on Bordet–Gengou agar (BD-Difco, 248200) supplemented with 15% defibrinated horse blood (Oxoid, SR0050C), and sub-cultured in the same medium for 24 h. Bacteria were suspended in physiologic salt to reach OD$_{650}$ of 1, and 400 µL of the suspension were pelleted. Pellets were re-suspended in 100 µL of PBS 1X (Gibco, 70013-016), 100 µL of lysis buffer (Roche, 04659180001), and 40 µL of proteinase K (Roche, 03115828001); heated at 65 °C for 10 min and at 95 °C for 10 min; and used for DNA extraction. Whole genome sequencing was performed using a NextSeq-500 system (Illumina, USA) at the Mutualized Platform for Microbiology of Institut Pasteur. For de novo assembly, paired-ends reads were clipped and trimmed with AlienTrimmer[50], corrected with Musket[51], merged (if needed) with FLASH[52], and subjected to a digital normalization procedure with khmer[53]. For each sample, remaining processed reads were assembled and scaffolded with SPAdes[54].

**Development of pan-genus *Bordetella* cgMLST scheme.** A genus-wide core genome MLST scheme[55], called *Bordetella* cgMLST v1.0, was established alongside a genus-wide locus annotation system (BORD loci), using the principles published for the genus *Neisseria*[56]. Six finished and annotated *Bordetella* genomes were uploaded into the BIGSdb Oxford database[57]: *Bordetella pertussis* Tohama I[22]; *Bordetella pertussis* CS[58]; *Bordetella bronchiseptica* RB50[22]; *Bordetella parapertussis* 12822[22]; *Bordetella petri* 12804[17]; and, *Bordetella avium* 197N[59]. Using the

GenomeComparator module of BIGSdb, the greatest number of common genes (1469) was found when *B. bronchiseptica* RB50 was used as the reference genome (even though *B. petri* 12804 had the greatest number of annotated loci, 5023). Genes described as 'hypothetical' or 'putative' were removed, resulting in 1415 loci. All loci entered into the database were given a BORD number of the format BORD000000, where the last six digits corresponded to the BB0000 numbers used in the annotation of *B. bronchiseptica* RB50[22]. Consequently, loci in this scheme are in the same order as they appear in the reference genome of *B. bronchiseptica* RB50. Other annotations and known aliases were included in the locus descriptions for comparison. As with other genus-wide locus schemes, this scheme can be expanded simply by adding additional loci, to give an expandable catalogue of the genus pangenome. We computed the fraction of genome length represented in the cgMLST scheme, by dividing the sum of allele lengths of the called loci, by the genome length, and averaged over all genomes per species (Supplementary Data 2).

**Merging of the Oxford and Pasteur databases**. Two BIGSdb databases were originally designed separately for distinct purposes: while Oxford's PubMLST database offered MLST, *Bordetella* cgMLST v1.0 (see above) and virulence genes schemes, the Pasteur database was designed for the sole initial purpose of *Bordetella pertussis* genotyping, with a Bp-specific cgMLST scheme and a Bp-virulence genes schemes[30].

To merge the data available in the two databases, we proceeded as follows. As per BIGSdb dual design, isolates genomes and provenance data were imported into the "isolates" database, whereas allelic definitions of MLST, cgMLST, and virulence genes were imported into the "seqdef" database.

Regarding the isolates database, we first dumped the Oxford database and uploaded it on the Institut Pasteur server. Second, we imported Pasteur isolates into this new database. To facilitate the understanding of historical origins of each entry, isolates identification numbers (BIGSdb ID number) were defined as follows: isolates from the original Oxford database were numbered from 1 to 1914 as in the original database (i.e., their identification numbers were untouched). Second, the isolates from the original Pasteur database were numbered from 10,000 to 12,214 (original Pasteur identification number + 10,000). We also completed the comment fields for these isolates, with the old identification number being added with the suffix word "Pasteur". We added "putative duplicate" in this comment field if the corresponding isolate name was present in both original databases. We also added a "duplicate number" field in the isolates database. If two or more entries were identified as corresponding to the same isolate, they were attributed the same duplicate number, consecutively. Duplicated data identification was based on a combined analysis of metadata and phylogeny tree reconstruction.

At time of merging and closure, the Oxford database comprised 1914 isolates entries, 57 of which were private, whereas the Pasteur database comprised 2180 entries, 2009 of which were private. As of January 7, 2022, the platform resulting from the merger comprises 2581 public isolates entries, and 4853 isolates in total when considering private entries.

**cgMLST schemes**. Two core genome MLST schemes are available. The first scheme, *Bordetella* cgMLST v1.0 (hereafter referred to as *cgMLST_genus)*, was initially defined and hosted in the Oxford platform and was designed to be applicable for the entire *Bordetella* genus. The second scheme, called *cgMLST_pertussis*, was originally hosted in the Pasteur database and was built for *B. pertussis* isolates genotyping only[30]. Note that only the latter scheme has attached cgST definitions, i.e., unique cgST identifiers are attached to each distinct allelic profile. Both cgMLST schemes are available in the merged resource.

**Genotyping schemes for vaccine antigens and virulence-associated genes**. In both original databases, virulence-associated gene schemes had been defined separately. We choose to keep all loci from the virulence scheme designed in Oxford and to add some additional relevant loci. These loci were then grouped into different schemes comprising either Bp vaccine antigens (*fim2, fim3, ptxP, ptxA, ptxB, ptxC, ptxD, ptxE, fhaB−2400_5550*), T3SS genes (*bopB, bopD, bopN, bsp22, bteA*), autotransporters (*bapC, brkA, tcfA, prn, vag8*), other toxins (*cyaA, dnt*) or phase biology genes (*bipA, bvgA, bvgS*). We decided to include the *prn* locus in the autotransporter scheme rather than in the *Bp* vaccine antigens scheme, because prn-negative isolates would not allow the definition of sequence types (ST) for the vaccine antigens scheme.

To make analyses of virulence-associated and vaccine antigen genes more user-friendly, common gene names used in the literature were used as locus identifier, instead of the locus tags that were initially used in the Oxford database (e.g., BORD005020 was replaced by *fim2* and BORD005021 by fim3). In addition, for five loci common to the original Pasteur and Oxford databases (i.e. *fim2, fim3, ptxA, ptxP,* and *tcfA*), the allele numbering was re-defined so that the alleles of each locus would match the nomenclature found in the literature for *B. pertussis* alleles; the correspondence between the original allele numbering system and the new one is defined in Supplementary Data 5. Subsequently, the numbering of alleles was simply incremented as new genomes scans led to novel allele definitions.

**Macrolide resistance**. The 23S rRNA allele table was incremented with a sequence defined as allele 13, which carries the only mutation described so far as being associated with macrolide resistance[41]. To facilitate the detection of putative macrolide resistant isolates and their lineage[40,41], we built a scheme named "macrolide resistance" that includes the following loci: 23S_rRNA, fhaB (full) and four *fhaB* fragments. As the *fhaB* locus, which has a full-length size of 10,773 bp, is often fragmented in the genomic assemblies, three smaller fragments were designed to cover evenly the *fhaB* sequence: fhaBx-1_3193, fhaBy-3190_7183, fhaBz-7180_10773; the ranges in these locus names correspond to the begin and end positions of the three fragments. A fourth fragment was designed to cover the region comprising 2 SNPs present individually in *fhaB2* and *fhaB3*[25]: fhaB-2400_5550. The allele numbering for *fhaB* fragments was defined so that the alleles of each locus would match the nomenclature found in the literature, i.e., alleles *fhaB1* and *fhaB2* correspond to those defined by van Loo[60], and *fhaB3* to the mutation C53301T[25]. Allele *fhaB3* of full size locus and restricted locus (fhaBy-3190_7183 and fhaB-2400_5550) is associated with *ptxP1* isolates originating from China which exhibit a macrolide resistance phenotype[25].

**Genome scanning for defined loci**. Reference genomes were selected from either type strain, reference genomes from literature and/or RefSeq genomes or most complete genomes available for each species (e.g., type strain 18323 and reference strain Tomaha for *B. pertussis*). All these genomes are grouped into the public project "Bordetella Genus Phylogeny" (project i.d. 23). We selected two representatives for each species, when available. All these genomes are found in the "Bordetella phylogeny" public project in the BIGSdb genomic platform. For the *cgMLST_genus* scheme, missing alleles were captured by relaxing the scanning parameters to 70% identity and 70% alignment to capture alleles from all *Bordetella* species, using the reference strains. New captured alleles were defined as type alleles. Then, all future detection of new alleles were based on 90% identity, 90% alignment parameters using defined type alleles as queries. Criteria for loci associated with virulence were stricter: only new alleles sharing 90% identity, 90% alignment with a type allele and with a complete CDS were recorded (excluding the *ptxP* promoter locus and the *fhaB* fragment fhaB-2400_5500). These strict parameters were used to minimize the detection of paralogous sequences and to exclude decaying alleles.

For each species, we defined reference strains taken either from literature or from the NCBI RefSeq database (Supplementary Data 4). Type alleles were defined from each reference isolate in order to constrain the search space so as to ensure future consistency of allele definitions. The definition of new alleles was subject to constraining parameters. To detect alleles, BLASTN thresholds were set to 90% of identity, 90% of alignment length coverage, and considering only the complete coding sequences (except for the *ptxP* locus promoter and the 23S rRNA gene, which are not protein CDSs). These strict parameters were used to minimize the detection of paralogous sequences.

**Phylogenetic analyses**. Phylogenetic analyses were performed based on concatenated alignments of individual gene loci defined in the database. To derive a phylogenetic tree based on cgMLST loci, we extracted the amino acid allele sequences of each locus and aligned them with MAFFT v7.467[61]. We then back-translated multiple amino acid sequence alignments to codon alignments and then concatenated them into a supermatrix. We used IQ-TREE v2.0.6[62] to infer a phylogenetic tree from this supermatrix of characters with a GTR + G + I evolutionary model and Gubbins (2.4.1, default parameters)[62,63]. We assessed branch supports with bootstrap (1000 replicates) and aLRT-SH methods[64]. Phylogenetic trees were drawn and annotated using iTOL[43]. When running Gubbins with cgMLST_genus data, alignments were concatenated following the loci order of the reference genome RB50.

To include the breadth of currently sampled *Bordetella* diversity, we first downloaded all genomic sequences from public databases and ran a rapid distance *mash*-based phylogenetic analysis (using Gklust software[65]). From this, we retained all representative genomes of unique deep branches and down-sampled shallow branches that were represented multiple times. The resulting dataset included the putative novel species initially described as genogroups by Spilker et al. on the basis of *nrdA* gene sequences[20]. The *Bordetella* diversity dataset is publicly available from the genomic platform as project *Bordetella* genus phylogeny (project i.d.: 23; https://bigsdb.pasteur.fr/cgi-bin/bigsdb/bigsdb.pl?db=pubmlst_bordetella_isolates&page=query&project_list=23&submit=1).

Genomic species were defined as groups of isolates that can be differentiated from other such groups using the complete genomic sequence-based ANI cutoff value of at 95%, which corresponds to a classically used threshold value for species delineation[31,66]. ANI values were computed for every possible pairwise comparison using fastANI v 1.33[67].

To analyze the phylogenetic structure within the BbGS, we selected 209 isolates: 181 *B. bronchiseptica* representative of the clonal complexes I and IV[1] and 9 others that did not belong to these complexes. This dataset was completed with isolates from *B. pertussis* (*n* = 11) and *B. parapertussis* (*n* = 10), selected to represent the known diversity within these two taxa: for Bp, lineage I (isolate 18323), lineage IIa (isolate B0887) and lineage IIb (which includes *ptxP1* isolate Tohama, and *ptxP3* isolates)[24]; and for *B. parapertussis*, the ovine lineage (isolate Bpp5) and the human lineage reference strain 12822. A public project encompassing all selected genomes is available in the BIGSdb genomic platform (project name: *B. bronchiseptica*

phylogeny; project i.d.: 24; https://bigsdb.pasteur.fr/cgi-bin/bigsdb/bigsdb.pl?db=pubmlst_bordetella_isolates&page=query&project_list=24&submit=1).

In the same way, 124*Bp* isolates selected to represent *Bp* genomic diversity were used to analyze the phylogenetic position of macrolide-resistant isolates. To this aim, we selected isolates representative of main *ptxP* branches (*ptxP* alleles 1, 2, 3, 19, and 21) and all genomes of isolates known to be resistant to erythromycin (*n* = 51). The dataset is publicly available from the genomic resource as project *B. pertussis* phylogeny (project i.d. 25; https://bigsdb.pasteur.fr/cgi-bin/bigsdb/bigsdb.pl?db=pubmlst_bordetella_isolates&page=query&project_list=25&submit=1).

**Reporting summary**. Further information on research design is available in the Nature Research Reporting Summary linked to this article.

## Data availability
BIGSdb Pasteur: https://bigsdb.pasteur.fr BIGSdb Pasteur, Bordetella resource: https://bigsdb.pasteur.fr/bordetella/ The sequence reads data generated in this study for *Bordetella bronchiseptica* isolates from France, and for *B. tumulicola*, *B. muralis*, and *B. tumbae* type strains, have been deposited in the European Nucleotide Archive (ENA) database, part of INSDC (NCBI/ENA/DDBJ), and is accessible under BioProject number PRJEB49946. *Bordetella* genomes list and accession numbers: Supplementary Data 4 *Bordetella* genus phylogeny dataset (92 isolates), project i.d. 23: https://bigsdb.pasteur.fr/cgi-bin/bigsdb/bigsdb.pl?db=pubmlst_bordetella_isolates&page=query&project_list=23&submit=1 *B. bronchiseptica* phylogeny dataset (211 isolates), project i.d. 24: https://bigsdb.pasteur.fr/cgi-bin/bigsdb/bigsdb.pl?db=pubmlst_bordetella_isolates&page=query&project_list=24&submit=1 *B. pertussis* phylogeny (124 isolates), project i.d. 25: https://bigsdb.pasteur.fr/cgi-bin/bigsdb/bigsdb.pl?db=pubmlst_bordetella_isolates&page=query&project_list=25&submit=1 *Bordetella* public genomes dataset (2085 isolates), project i.d. 27: https://bigsdb.pasteur.fr/cgi-bin/bigsdb/bigsdb.pl?db=pubmlst_bordetella_isolates&page=query&project_list=27&submit=1 *Bordetella* genus nrdA dataset (180 isolates), project i.d. 29: https://bigsdb.pasteur.fr/cgi-bin/bigsdb/bigsdb.pl?db=pubmlst_bordetella_isolates&page=query&project_list=29&submit=1 iTOL interactive trees: https://itol.embl.de/shared/1l7Fw0AvKOoCF Source data are provided with this paper.

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

## Acknowledgements

We thank Julien Guglielmini and Alexis Criscuolo (Bioinformatics and Biostatistics HUB in Institut Pasteur) for help with the BIGSdb-Pasteur database configuration and for advice regarding bioinformatics analyses. This work was supported financially by the French Government's Investissement d'Avenir program Laboratoire d'Excellence "Integrative Biology of Emerging Infectious Diseases" (ANR-10-LABX-62-IBEID). BIGSdb development is funded by a Wellcome Trust Biomedical Resource grant (218205/Z/19/Z). This work used the computational and storage services provided by the IT Department at Institut Pasteur. The National Reference Center for Whooping Cough and other Bordetella Infections is supported by Institut Pasteur and Santé publique France (Public Health France).

## Author contributions

Conceptualization, Supervision: S. Brisse designed and coordinated the study. Methodology, software, validation, and data curation: S. Bridel performed the merging the two databases into BIGSdb-Pasteur and the phylogenetic analyses, as well as extensive databases curation (metadata and duplicated data cleaning). V.B. performed analyses of virulence and antigen loci sequences and established the literature-based allelic nomenclature. S.H., K.A.J., and M.C.J.M. created the *Bordetella* cgMLST v1.0 scheme. K.A.J. and M.C.J.M. provided the data from BIGSdb Oxford and developments of the BIGSdb platform. N.A., A.L., S.G., J.T., and E.M. analyzed *Bordetella* isolates at the French NRC and performed genomic sequencing. B.B. provided support for the BIGSdb-Pasteur platform deployment and maintenance. Writing-original draft preparation, review and editing: S. Bridel, V.B., and S. Brisse wrote the original draft. All authors contributed to and approved the final version of the manuscript.

## Competing interests

The authors declare no competing interests.
