## [Peer Review File · Nature Communications]

Reviewers' Comments:

Reviewer #1:

Remarks to the Author:

Bridel et al. have combined and harmonized two existing allele databases for the genomic analyses of *Bordetella* species isolates. Due to the unique taxonomic composition of this genus, there is clear value in efforts to ensure consistent nomenclature between the more medically focal species *B. pertussis* and the species at varied phylogenetic distances. Importantly, the authors have made their resulting schemes readily available to the public. I have the following comments regarding the utility of the schemes and their presented evaluation.

[1] It is unclear how such a unified database of genus-wide core genes can facilitate analyses of phenotypic diversity with the genus as the authors have suggested (Ln 98). Rather, it is often the content of accessory genes which frequently defines phenotypic differences among divergent species and such analyses have been previously reported for *Bordetellae* (Linz et al. 2016). The presented work suggests allele profiles of the T3SS may discriminate the BbGS ("classic *bordetellae*"). Can the authors provide any similar examples to demonstrate the power of the scheme beyond the narrow scope of this "genomic species"?

[2] A genus-wide cgMLST scheme is expected to capture a smaller percentage of total nucleotides for those species with larger genomes or with more variable gene content. The authors suggest that the unified genus database can be used to track populations of individual species but it appears that the design may effectively reduce the resolution of their analyses, particularly within individual species. In addition to indicating the number of uncalled loci (Figure S3), it would be informative to also report the queried genome fraction for each species and some corresponding quantification of genomic resolution provided by the scheme.

[3] Can the authors provide an independent phylogenetic reconstruction to corroborate the accuracy of their genus cgMLST scheme for capturing evolutionary relationships presented in Figure 1?

Minor comments:

Ln 112. How does the unified nomenclature facilitate epidemiologic study of *B. pertussis* beyond previous report of the Pasteur cgMLST scheme (Bouchez et al. 2018)?

Ln 183. The widely accepted species boundary is ANI <95%. Why do the authors suggest this group should be investigated as a potential new species if the ANI values are above that threshold?

Ln 214-221. The distribution and frequency of T3STs within each sublineage appears to reflect the number of included isolates and the within-group sequence diversity. Is this not expected?

Ln 242-244. The Bp genome encodes 3 copies of 23S, how are these alleles defined? Do they require or assume homozygous/identical 23S sequences in all 3 copies? If so, how can that be accurately distinguished from resistant mutation to only 1 or 2 copy?

Ln 314. Evidence of divergent *B. bronchiseptica* was previously reported in Weigand et al. 2019 and the isolate sequence (2142 I328) is included here in the described Bbs lineage II.

Ln 353-358. Is this sampling limitation expected to impact the gene/locus content of the cgMLST schemes, the catalog of observed alleles, or both?

Fig 2 – Leaf node symbols and colors are very difficult to distinguish. Consider annotating these in another way, such as additional ring(s) of color bars for example.

Fig 2 and Fig S2 – The text suggests the reconstruction includes Bp and Bpp, and is presented as evidence for polyphyletic origin of Bpp, but the visualized tree appears to only contain Bb nodes? Headings and figure titles also suggest emphasis is on Bb. Please check for errors.

Reviewer #2:

Remarks to the Author:

The manuscript by Bridel and Bouchez et al is a comprehensive genomic analysis of the *Bordetella* genus. It describes the merged dataset from two online databases, in addition to newly-sequenced

type strains for some species. The overall collection is a very valuable resource, and the authors identify potentially novel species. However there are some issues with the analysis, in terms of the methodology, the description of the population (including understanding which observations are novel), and the epidemiological interpretation.

Major criticisms:

Point A: Description of the population – this is very vague throughout the paper, making it very difficult to keep track of how different subdivisions of the population relate to one another.

Additional analysis and more specific language would be greatly beneficial. For instance:

A.1: the authors state, "The nomenclature of genotypic markers, sublineages and strain subtypes needs unification to facilitate collective studies of the global epidemiology and population dynamics in Bp." This seems like an ideal opportunity to develop and validate such a nomenclature. The authors define a cgMLST scheme. However, the resolution of such typing is too high to allow for an intuitive description of the population, hence the authors resort to poorly-defined terms such as "lineage", "genomic species" and "genogroups" to analyse the population in the text and figures. There are many methods available for clustering genomic datasets, like Hierarchical Clustering of CgMLST (<https://enterobase.readthedocs.io/en/latest/features/clustering.html>), Poppunk (Lees et al <https://genome.cshlp.org/content/29/2/304.full>) and SNV genotyping (Hawkey et al <https://www.nature.com/articles/s41467-021-22700-4>). The authors should decide on a suitable method for systematically dividing the population into useful units, and use these to describe the population structure using a consistent terminology that can be applied to all isolates in the collection.

A.2: In the Abstract, the authors state:

"the three novel species *B. tumulicola*, *B. muralis* and *B. tumbae* in a clade with *B. petrii* and revealed 18 yet undescribed species."

Based on Table S2 and Figure S1, it appears most of these correspond to genogroups identified as candidate new species by Spilker et al, which should be acknowledged. It would be helpful for the authors to include a table listing the actual and candidate type strains for each proposed species in the dataset, along with the relevant accession codes.

A.3: the Abstract and main text refers to "genogroups", but these are not defined, and their origin is not described in the Introduction. They appear to be based on a previous publication, and it is not clear how many of the isolates in this publication could be assigned to genogroups. This information should be provided, and compared with any systematic analysis of the population added to this manuscript.

Point B: Dataset and analysis methodology – there are some problems with making data easily available to others, and the methods used to analyse it in this paper:

B.1: The authors state, "The phylogenetic tree was obtained based on the concatenated multiple sequence alignments of the 1,415 core gene sequences from the cgMLST_genus scheme; recombination was accounted for using Gubbins." However, software such as ClonalFrameML and Gubbins runs on whole genome alignments, not concatenated gene alignments, as they depend on the spatial arrangement of mutations. The Gubbins authors address this issue here:

<https://github.com/sanger-pathogens/Roary/issues/267>.

B.2: The authors state, "As of January 7th, 2022, the platform resulting from the merger comprises 2,581 public isolates entries, and 4,853 isolates in total when considering private entries." As the database will soon be expanded from the set analysed in this paper, it would be helpful if the authors could provide the full set analysed in this study, so it can be reproduced, or re-analysed. Table S4 does not contain 2,581 rows, as would be expected of such a table. Also, this table should be a spreadsheet, not a multi-page PDF.

B.3: The cited Weigand et al wgMLST paper contains informative visualizations of complete datasets using unrooted distance-based trees. I understand rooted trees of >2,000 isolates are not likely to be informative. However, the <https://bigsd.b.pasteur.fr> site must offer some way of visualizing large datasets. It would be very helpful if the authors could demonstrate how best to visualize the overall contents of the database.

Point C: Epidemiological interpretation – this could be improved for the general audience of the journal. Much of the detail of the last two sections of the Results could be moved to the supplementary text, as it is not of such broad interest as the first parts of the Results. This would

make way for addressing important questions like -

C.1: The authors state in the Introduction, "the population dynamics within Bp are an important topic of epidemiological surveillance, in light of vaccine-escape evolution and the possible emergence and global dissemination of antimicrobial resistance". The term "vaccine escape" also features in the first line of the Abstract, and in the Conclusions. However, the term features nowhere in the Results. Are the authors able to give an overview of -

C.1.1: any differences in population structure between countries using acellular and whole-cell vaccines

C.1.2: how common are the vaccine escape mutants and how far have such strains spread? The authors refer to "ptxP3 branches" and "ptxP3 strains", but these are not labelled on the figures, and not explained in the text.

C.1.3: any evidence of horizontal transfer of these mutations? The absence of horizontal transfer would be interesting in itself

C.2: It would be helpful to understand the distribution of sequence types between countries to a greater extent. Is the observed confinement of the resistant lineage to China unusual when compared to the geographic range of sensitive lineages?

C.3: The authors state, "The remaining lineage II isolates were six human clinical isolates from France, collected from adults (mean age = 71.7 years) displaying pulmonary infections. These observations clearly establish the pathogenic potential of Bbs lineage II.". Isolation from a sick individual does not establish these bacteria as pathogens, unless the collections were from the bloodstream. The authors should modify this statement accordingly.

C.4: The authors state, "The independent origins of Bpphu and Bppov have been debated". They describe the place of Bppov phylogeny, but they provide no interpretation of which side of the debate they agree with. In the Discussion they state, "More genomes of Bppov would be needed". It is not clear how informative this part of the manuscript is without more interpretation or data.

Minor criticisms:

- References 1 and 5 are duplicates

- The Introduction starts by stating "Bordetellae are beta-proteobacteria that can be found in the environment", but then most of the described species are human- or animal-restricted, which I found confusing

- Unclear what is meant by "suboptimal services to the user's community"

- Italicisation needed of "fhaB3" (Discussion) and "fim2" (Abstract)

- Do the authors use "isolate" and "strain" interchangeably? They should clarify if this is the case, or select one over the other

**Reviewer #1 (Remarks to the Author):**

*Bridel et al. have combined and harmonized two existing allele databases for the genomic*
*analyses of Bordetella species isolates. Due to the unique taxonomic composition of this*
*genus, there is clear value in efforts to ensure consistent nomenclature between the more*
*medically focal species B. pertussis and the species at varied phylogenetic distances.*
*Importantly, the authors have made their resulting schemes readily available to the public. I*
*have the following comments regarding the utility of the schemes and their presented*
*evaluation.*

**Our answer:** thank you for the positive comments.

*[1] It is unclear how such a unified database of **genus-wide core genes** can facilitate analyses*
*of **phenotypic diversity with the genus** as the authors have suggested (Ln 98). Rather, it is*
*often the content of accessory genes which frequently defines phenotypic differences among*
*divergent species and such analyses have been **previously reported** for Bordetellae (Linz et al.*
*2016)*

**Our answer:** We agree that acquisition and loss of accessory genes are the most
relevant for phenotypic variation. In our platform, two different typing schemes are based on
core genes (cgMLST_genus and cgMLST_pertussis), but many accessory genes are also
defined as loci, e.g., T3SS genes, or antigen/virulence genes of *B. pertussis*. Using our library
and BIGSdb interface, core genome-based analysis can easily be completed with specific
studies of accessory genes variation.

*The presented work suggests **allele profiles of the T3SS may discriminate the BbGS***
*(“classic bordetellae”). Can the authors provide any **similar examples** to demonstrate the*
*power of the scheme beyond the narrow scope of this “genomic species”?*

**Our answer:** In fact, the T3SS scheme is meant for this species only. It is a
characteristic of multi-species libraries that some gene loci or schemes will only be relevant
for specific species. Another example is the agST scheme, also used to the BbGS members
(Figure S5). We have not yet developed schemes for the other species but this possibility
exists for future studies. We now discuss this limitation (Discussion, lines 400-405).

[2] A **genus-wide cgMLST scheme** is expected to capture a smaller percentage of total
nucleotides for those species with larger genomes or with more variable gene content. The
authors suggest that the unified genus database can be used to track populations of individual
species but it appears that the design may effectively reducing the resolution of their analyses,
particularly within individual species. In addition to indicating the number of uncalled loci
(Figure S3), it would be informative to also report the **queried genome fraction** for each
species and some corresponding quantification of **genomic resolution** provided by the
scheme.

**Our answer:** We thank the reviewer for this suggestion. We have added the
information on the percentage of genome length covered by the genus-wide cgMLST scheme,
in a supplementary table (**Table S3**). Core genome schemes with a few hundred genes are
already highly discriminatory for population biology and lineage classification purposes. The
genus-wide scheme, based on 1,415 loci, represents a good initial genotyping approach. We
agree that more comprehensive schemes or typing approaches (such as whole genome SNP)
would be more discriminatory, but at the expense of ad-hoc developments for each
species/sublineage. We discuss the issue of limited discrimination (lines 329-340).

[3] Can the authors provide an **independent phylogenetic reconstruction** to corroborate the
accuracy of their genus cgMLST scheme for capturing evolutionary relationships presented in
Figure 1?

**Our answer:** An independent phylogenetic reconstruction, based on whole genome
analysis was provided in **Figure S4** (now **Figure S2**), which presents a K-mer distance-based
phylogenetic tree obtained using JolyTree (<https://gitlab.pasteur.fr/GIPhy/JolyTree>), with
branch support values.

*Minor comments:*

Ln 112. How does the unified nomenclature facilitate epidemiologic study of *B. pertussis*
beyond previous report of the Pasteur cgMLST scheme (Bouchez et al. 2018)?

**Our answer:** The previously developed *B. pertussis* cgMLST scheme (Bouchez et al.
2018) was designed to maximize subtyping resolution power; allele definitions were provided
and cgST are defined based on that scheme. But there was no nomenclature for higher level
groups (for example, groups of cgMLST profiles differing by up to 10, 25, or 100
mismatches). In the present work we provide, with the Bp_vaccine antigens scheme and
attached sequence types, a *B. pertussis* sublineage nomenclature. Besides, this scheme
provides the advantage that it is based on loci that are commonly used in the literature (*ptxP*,
*ptxA*, *fim2* or *fim3*...), so that the alleles themselves have meaning. And it marks important
subdivisions of the *B. pertussis* tree (Figure 3). Therefore, we do believe this scheme will
facilitate communication within the *B. pertussis* epidemiology and population biology
community.

*Ln 183. The widely accepted species boundary is ANI <95%. Why do the authors suggest this*
*group should be investigated as a potential new species if the ANI values are above that*
*threshold?*

**Our answer:** ANI 94-96% is a usually accepted range and is used only as a guide for
species boundaries definitions (Chirag et al.: [https://www.nature.com/articles/s41467-018-](https://www.nature.com/articles/s41467-018-07641-9)
[07641-9](https://www.nature.com/articles/s41467-018-07641-9) ; Konstatinidis et al: <https://www.pnas.org/doi/full/10.1073/pnas.0409727102>;
[Richter and Rosselo-Mora https://www.pnas.org/doi/full/10.1073/pnas.0906412106](https://www.pnas.org/doi/full/10.1073/pnas.0906412106)), and the
interpretation of ANI data for species boundary definition should be considered together with
evolutionary relationships
(<https://www.microbiologyresearch.org/content/journal/ijsem/10.1099/ijsem.0.004124>).

Hence, the threshold used to define particular species remains largely flexible, using this
range as a guide. Some species may be defined with slightly lower thresholds and other with
higher ones (e.g., 96%), depending on phylogenic structure and the genetic heterogeneity of
particular species. Considering that the three taxonomic species *B. bronchiseptica*, *B.*
*pertussis* and *B. parapertussis* belong to a single genomic species, and show 97.8–98.7% ANI
between each other, it could be a logical extension to consider *B. bronchiseptica* lineage II as
taxonomically distinct, even if its ANI is >95% with the three other taxa. Note that we
remained open on the relevance of such a proposal and did not propose a taxonomic update
here.

*Ln 214-221. The distribution and frequency of T3STs within each sublineage appears to*
*reflect the number of included isolates and the within-group sequence diversity. Is this not*
*expected?*

**Our answer:** Yes, this was expected; however, the main message of this paragraph
was to underline how Bbs sublineages are characterized by unique T3ST sequence types. We
have rephrased the sentence for more clarity [Line 219-227].

*Ln 242-244. The Bp genome encodes 3 copies of 23S, how are these alleles defined? Do they*
*require or assume homozygous/identical 23S sequences in all 3 copies? If so, how can that be*
*accurately distinguished from resistant mutation to only 1 or 2 copy?*

**Our answer:** The reviewer raises a highly pertinent point. Erythromycin-resistant
strains of *B. pertussis* could display either homozygous or heterozygous *rrn* operons with
either one, two or three copies of the rRNA having the A2047G mutation (Bartkus *et al.*
2003). Our platform uses assembled sequences, which are derived from the majority
consensus, erasing the heterogeneity if any. It is not an adequate tool to detect heterogeneity,
instead, the heterogeneity within read sets should be investigated. In our dataset, all *B.*
*pertussis* isolates known to be resistant for erythromycin were found with 23S_rRNA allele
13 instead of 1, and we simply infer that at least two of the three copies were mutated when
allele 13 is found. According to Feng *et al.* 2021, a mutation in a single copy is already
sufficient to raise the MIC to > 64 µg/mL; thus, macrolide-resistant strains would be missed
using consensus assemblies. We have added this limitation [**Discussion, §§ 392-354; Lines**
**405-413**].

*Ln 314. Evidence of divergent B. bronchiseptica was previously reported in Weigand et al.*
*2019 and the isolate sequence (2142 I328) is included here in the described Bbs lineage II.*

**Our answer:** The reviewer is right; thank you for pointing this out; we have added the
reference Weigand *et al.* (2019) and referred to it in the discussion [**Lines 352-354**] and
mention strain I328 in the abstract too [Line 55].

*Ln 353-358. Is this sampling limitation expected to impact the gene/locus content of the*
*cgMLST schemes, the catalog of observed alleles, or both?*

**Our answer:** The lack of inclusion of genome sequences from the underrepresented
world region probably restricts the recorded allelic diversity, as there is high allelic diversity
within the *ptxPI* branches (Bart *et al.* 2014; Bouchez *et al.* 2018). However, isolates in our
dataset collected in high income countries such as the USA or Europe during the whole cell
vaccine era, partly fill this gap. Hence, this imbalance is not expected to impact the gene locus
content of the core genome, as lineages of the *ptxPI* branch were included in the definition of
the cgMLST scheme, and as there is little gene content variation among Bp genomes. We now
highlight the sequence imbalance across countries [**§Discussion, §§ 392-354; Lines 392-400**].

*Fig 2 – Leaf node symbols and colors are very difficult to distinguish. Consider annotating*
*these in another way, such as additional ring(s) of color bars for example.*

**Our answer:** Thank you. We have modified the figure following the reviewer's suggestion;
Changes in the figure include:

- • A specific range of color for each lineage (I-1: magenta/pink/light purple, I-2: red, I-3:
blue, I-4: green/olive green and II: dark purple)
- • Leaf node symbols were change to pastel colors, to be more distinct from the outer
ring colors (T3ST). Host name's first letter was added into the symbol to facilitate
reading
- • For each lineage, if a T3ST is shared by several isolates within a given
lineage/sublineage, they share the same color (except for lineage II)
- • T3ST singletons are represented in gray, alternating 2 shades of grey when they are
adjacent.
- • Missing T3STs (due to missing data or absence of one or more of the loci) are
represented in white

*Fig 2 and Fig S2 – The text suggests the reconstruction includes Bp and Bpp, and is presented*
*as evidence for polyphyletic origin of Bpp, but the visualized tree appears to only contain Bb*
*nodes? Headings and figure titles also suggest emphasis is on Bb. Please check for errors.*

**Our answer:** In the header and title, we meant ‘*B. bronchiseptica* genomic species’
rather than *B. bronchiseptica*. The title of the figure was corrected. BbGS does comprise *B.*
*pertussis* and *B. parapertussis*, and Figures 2 and S2 do contain isolates from these species.
We renamed figure 2 “**Cladogram of the *Bordetella bronchiseptica* genomic species**” and
**Figure S3 « Phylogeny of the *B. bronchiseptica* genomic species»**. Both legends already
state the inclusion of *Bp* and *Bpp*.

**Reviewer #2 (Remarks to the Author):**

*The manuscript by Bridel and Bouchez et al is a comprehensive genomic analysis of the*
*Bordetella* *genus. It describes the merged dataset from two online databases, in addition to*
*newly-sequenced type strains for some species. The overall collection is a very valuable*
*resource, and the authors identify potentially novel species. However there are some issues*
*with the analysis, in terms of the methodology, the description of the population (including*
*understanding which observations are novel), and the epidemiological interpretation.*

**Our answer:** thank you for your comments and general appreciation.

*Major criticisms:*

*Point A: Description of the population – this is very vague throughout the paper, making it*
*very difficult to keep track of how different subdivisions of the population relate to one*
*another. Additional analysis and more specific language would be greatly beneficial. For*
*instance:*

*A.1: the authors state, “The nomenclature of **genotypic markers, sublineages and strain***
***subtypes** needs **unification** to facilitate collective studies of the global epidemiology and*
*population dynamics in *Bp*.” This seems like an ideal opportunity to develop and validate*
*such a nomenclature.*

*The authors define a cgMLST scheme. However, the resolution of such typing is too high to*
*allow for an intuitive description of the population, hence the authors resort to poorly-defined*
*terms such as “**lineage**”, “**genomic species**” and “**genogroups**” to analyse the population in*

*the text and figures. There are many methods available for clustering genomic datasets, like*
*Hierarchical Clustering of cgMLST*
*(<https://enterobase.readthedocs.io/en/latest/features/clustering.html>), Poppunk (Lees et al*
*<https://genome.cshlp.org/content/29/2/304.full>) and SNV genotyping (Hawkey et al*
*<https://www.nature.com/articles/s41467-021-22700-4>). The authors should decide on a*
*suitable method for systematically dividing the population into useful units and use these to*
*describe the population structure using a **consistent terminology** that can be applied to all*
*isolates in the collection.*

**Our answer:** Thank you for the comments. We agree that it would be interesting to
define an all-encompassing genomic nomenclature in this work. We are aware of the interest
of such nomenclatures and have done so recently for *Klebsiella* for example
(<https://www.biorxiv.org/content/10.1101/2021.07.26.453808v1>). However, nomenclatures
based on genomic phylogenetic structures of populations within species require many more
genomes than currently available for each individual *Bordetella* species, except for *B.*
*pertussis*. But in this species, phylogenetic subdivisions are subtle as they typically rely on
single SNPs. Here, a nomenclature of major subdivisions is instead proposed for *B. pertussis*
based on important landmark genetic markers (*ptxP3*, *fim*): agSTs. In the future, a cgMLST
based nomenclature could be proposed for other species, but large representative sets of
genomes will be needed to properly define the phylogenetic discontinuities and corresponding
distance cutoffs.

We would like to highlight that genogroups were defined in Spilker *et al.* (2014). Here we
used ANI to define genomic species (sometimes called genomospecies, but we prefer the
simpler term genomic species). During this revision, we added a table of correspondence of
these genomic species with previous denominations of genogroups (**Table 1**) to summarize
this. A paragraph was added under **§Methods, §§Phylogenetic analyses [Lines 587-589]** to
define the term “genomic species”. We also added some text to pinpoint our proposal to shift
from “genogroups” naming to “genomic species”, based on our results [**Lines 148-153**].

We have harmonized the usage of species, genomic species and lineage throughout.

*A.2: In the Abstract, the authors state: “the three novel species *B. tumulicola*, *B. muralis* and*
**B. tumbae* in a clade with *B. petrii* and revealed 18 yet undescribed species.”Based on Table*

*S2 and Figure S1, it appears most of these correspond to genogroups identified as candidate*
*new species by Spilker et al, which should be acknowledged. It would be helpful for the*
*authors to include a table listing the actual and candidate type strains for each proposed*
*species in the dataset, along with the relevant accession codes.*

**Our answer:** Reference to Spilker et al. (2014), where genogroups were defined, is
provided at several places in our manuscript. This reference is found in the main text (e.g Line
55, **Lines 352-354**). In addition, as requested, we now provide **Table 1 and Table S2** to
clarify correspondence among genomic species and genogroups, and indicate the reference
strains proposed for each undescribed species (taxonomic type strains would have to be
defined upon formal species descriptions). Note that in Spilker *et al* (2014), only 11
species/genogroups were present, and the phylogenetic tree was based on the single gene
*nrdA*. Using complete genomes, we found that genogroup 8 strains correspond in fact to 2
different genomic species whereas genogroup 6 corresponds to the divergent lineage inside
the *B. bronchiseptica* genomic species, which we named BbGS lineage II. We also found 8
additional genomic species we named *Bordetella* genomic species 19 to 26. We added these
denominations to Figure 1 and Figure S2. And we explain why the 20 genomic species are
number up to 26, as we avoided the use of ambiguous numbers (line 148-151 and footnote to
Table 1).

*A.3: the Abstract and main text refers to “genogroups”, but these are not defined, and their*
*origin is not described in the Introduction. They appear to be based on a previous*
*publication, and it is not clear how many of the isolates in this publication could be assigned*
*to genogroups. This information should be provided and compared with any systematic*
*analysis of the population added to this manuscript.*

**Our answer:** Indeed, “genogroup” was introduced by Spilker et al (2014). This was in
fact stated in our introduction; and the term ‘genomosp.’ was used by Spilker *and coll.* in the
NCBI sequence database. We have rephrased to make this clearer. We have also added **Table**
**1 and Table S2** to provide the detailed correspondence of ‘genogroups’ with genomic species
from our study, and a novel **Figure S8** based on *ndrA* gene sequences (see point below).
Thank you for the suggestion, as this indeed clarifies the matter.

*Point B: Dataset and analysis methodology – there are some problems with making data*
*easily available to others, and the methods used to analyse it in this paper:*

*B.1: The authors state, “The phylogenetic tree was obtained based on the concatenated*
*multiple sequence alignments of the 1,415 core gene sequences from the cgMLST_genus*
*scheme; recombination was accounted for using Gubbins.” However, software such as*
*ClonalFrameML and Gubbins runs on whole genome alignments, not concatenated gene*
*alignments, as they depend on the spatial arrangement of mutations. The Gubbins authors*
*address this issue here: <https://github.com/sanger-pathogens/Roary/issues/267>.*

**Our answer:** We acknowledge that Gubbins was developed for whole genome
alignments. However, as a first approximation, the concatenate of the cgMLST genes
sequences may be used as well, given that the chromosomal order is respected within the
concatenate (cgMLST_genus loci are ordered according to their position of the annotated
genome of *B. bronchiseptica* RB50 – and the same for *B. pertussis* scheme). Hence, regions
of high SNP density, especially those spanning several gene loci, would be detected as well
(arguably less precisely though, given that intergenic regions and missing genes are not
represented in the concatenate). The structure of the Gubbins-derived trees was in good
agreement with the whole genome phylogeny, both for the genus and for the BbGS (**Figures**
**S2 and S3**). We would therefore prefer to keep the Gubbins tree, and provide the alternative
whole genome trees in supplementary data.

*B.2: The authors state, “As of January 7th, 2022, the platform resulting from the merger*
*comprises 2,581 public isolates entries, and 4,853 isolates in total when considering private*
*entries.” As the database will soon be expanded from the set analysed in this paper, it would*
*be helpful if the authors could provide the full set analysed in this study, so it can be*
*reproduced, or re-analysed. Table S4 does not contain 2,581 rows, as would be expected of*
*such a table. Also, this table should be a spreadsheet, not a multi-page PDF.*

**Our answer:**

The entire public dataset is available directly in the public database
(https://bigsd.b.pasteur.fr/cgi-bin/bigsd/bigsd.pl?db=pubmlst_bordetella_isolates).

Nevertheless, this public database contains some duplicated entries or incomplete ones (which

do not have associated genomes because they were entered at the time of 7-gene MLST), so
curation was needed before running genomic analyses. This is why we have created relevant
curated selections of isolates.

The curated selections of isolates used in this study are available within the database
as public projects ([https://bigsdb.pasteur.fr/cgi-
bin/bigsdb/bigsdb.pl?db=pubmlst_bordetella_isolates&page=projects](https://bigsdb.pasteur.fr/cgi-bin/bigsdb/bigsdb.pl?db=pubmlst_bordetella_isolates&page=projects)). These ‘projects’ are in
fact lists of BIGSdb IDs and can be used to reproduce results presented in this study. To
address the concern of the reviewer and provide a stable list in case the projects contents
would be modified by curators in the future, we have added to **Table S4**, a fixed list of all
isolates used in each of the three different projects. Thank you indeed for the suggestion.

*B.3: The cited Weigand et al wgMLST paper contains informative visualizations of complete*
*datasets using unrooted distance-based trees. I understand rooted trees of >2,000 isolates are*
*not likely to be informative. However, the <https://bigsdb.pasteur.fr> site must offer some way of*
*visualizing large datasets. It would be very helpful if the authors could demonstrate how best*
*to visualize the overall contents of the database.*

**Our answer:** The BIGSdb platform does contain a number of plugins allowing users
to build unrooted trees or networks (https://bigsdb.readthedocs.io/en/latest/data_analysis.html,
*e.g* Microreact or GrapeTree). Following the reviewer’s suggestion, we added illustrations
from two of the projects we defined and curated. First, we used all 1,924 genomes from public
databases (listed in BIGSdb project “Public Genomes”, ID 27) to produce a GrapeTree
visualization (**Figure S7**). This figure is focused only on BbGS genomes, as it represents 95%
of the public dataset, and as profile comparisons are meaningful only within species. Second,
we also propose in a novel **Figure S8**, an iTOL visualization obtained from *nrdA* locus
sequences for public isolates representative of the genus diversity; this figure provides the
additional benefit of comparing our genomic species definitions with the *ndrA*-based analysis,
which was used for initial genogroup definitions.

*Point C: Epidemiological interpretation – this could be improved for the general audience of*
*the journal. Much of the detail of the last two sections of the Results could be moved to the*

*supplementary text, as it is not of such broad interest as the first parts of the Results. This*
*would make way for addressing important questions like –*

*C.1: The authors state in the Introduction, “the population dynamics within Bp are an*
*important topic of epidemiological surveillance, in light of vaccine-escape evolution and the*
*possible emergence and global dissemination of antimicrobial resistance”. The term*
*“vaccine escape” also features in the first line of the Abstract, and in the Conclusions.*
*However, the term features nowhere in the Results.*

**Our answer:** Thank you for your comment. We have modified accordingly to better highlight
epidemiological aspects for the general audience. First, we have removed the term ‘vaccine
escape’ (we realize it is potentially misleading, as escape is only partial) and harmonized into
vaccine-driven evolution or partial vaccine escape. Partial vaccine-escape evolution of *B.*
*pertussis* is supported by the allelic divergence observed for vaccine antigens *ptxA* and *fim3* as
already described (Bart *et al.* 2014) and by the emergence of PRN-deficient isolates. Second,
to make our text more suitable for a general audience, we have modified extensively the
corresponding results section (**Lines 238-249**) and updated **Figure 3** accordingly. We also
clarified by adding (just after the above paragraph), a paragraph on pertactin, an important
component of vaccine-driven evolution. Figure 3 was updated, and now allows visualizing the
main allelic changes observed with identification of key nodes (*ptxA1*, *ptxP1*, *fim2-1*, *fim3-1*
and *fim3-2*).

*Are the authors able to give an overview of :*

*C.1.1: any differences in population structure between countries using acellular and whole-*
*cell vaccines*

**Our answer:** As was stated in Lines 353-358 of the initial submission, our dataset is
imbalanced towards genomic sequences from high-resource countries, which use acellular
vaccines, so it is hard to provide a strongly documented answer to this question. We discuss
this limitation in a ‘limitations’ paragraph at the end of the discussion. We have also added
**Figure S6** (to address another comment below), which partially answers the question, as
China (and Tunisia) has been using whole cell vaccines much more recently.

*C.1.2: how common are the vaccine escape mutants and how far have such strains spread?*
*The authors refer to “ptxP3 branches” and “ptxP3 strains”, but these are not labelled on the*
*figures, and not explained in the text.*

**Our answer:** *ptxP3* was labeled in Figure 3. In most countries using acellular
vaccines, *ptxP3* isolates have replaced *ptxP1* isolates since vaccine introduction (Bart *et al.*
2014). A notable exceptional situation is China, where *ptxP1* isolates still predominate (Xu *et*
*al.* 2019). The added **Figure S6** now illustrates this point, and the paragraph on the landmark
mutations in *B. pertussis* [**Lines 238-249**] should clarify this point too.

The vaccine antigen pertactin is not produced by an increasing part of recent
circulating strains. The *prn* locus is available in BigsDB database and was included in the
Autotransporters scheme. To address the reviewer comment, we have added a paragraph on
pertactin in the results section [**Lines 251-257**].

*C.1.3: any evidence of horizontal transfer of these mutations? The absence of horizontal*
*transfer would be interesting in itself*

**Our answer:** Horizontal gene transfer is considered rare or even absent within *B.*
*pertussis* or between *B. pertussis* and other taxa. More specifically, homoplasies in the
phylogenies were never observed for these mutations, indicating a lack of horizontal transfer.

*C.2: It would be helpful to understand the distribution of sequence types between countries to*
*a greater extent. Is the observed confinement of the resistant lineage to China unusual when*
*compared to the geographic range of sensitive lineages?*

**Our answer:** We thank reviewer for this comment. We added **Figure S6** to illustrate
how the resistant lineage bearing *fhaB3* allele is unusually restricted to China. We added
[§Results, Lines 245-249] the following sentence: “In our dataset, considering isolates
collected after 2008, *ptxP1* isolates mainly originated from China, whereas *ptxP3* isolates
were predominantly from France and the USA (**Figure S6**)”.

*C.3: The authors state, “The remaining lineage II isolates were six human clinical isolates*
*from France, collected from adults (mean age = 71.7 years) displaying pulmonary infections.*
*These observations clearly establish the pathogenic potential of Bbs lineage II.”. Isolation*
*from a sick individual does not establish these bacteria as pathogens, unless the collections*
*were from the bloodstream. The authors should modify this statement accordingly.*

**Our answer:** The word ‘potential’ that we had used left open the possibility of this
sublineage not being pathogenic; nevertheless, we agree that this was unclear, and removed
the sentence accordingly; we added this more neutral sentence in the discussion: “Defining
the pathogenic potential and epidemiology of *Bbs* lineage II is an important topic of future
research”.

*C.4: The authors state, “The independent origins of Bpphu and Bppov have been debated”.*
*They describe the place of Bppov phylogeny, but they provide no interpretation of which side*
*of the debate they agree with. In the Discussion they state, “More genomes of Bppov would be*
*needed”.* *It is not clear how informative this part of the manuscript is without more*
*interpretation or data.*

**Our answer:** We agree with the reviewer that this conveyed little information and
have removed entirely the mentioned paragraph. The only genome available was nevertheless
kept in the phylogenetic analysis for information purpose. We now simply discuss the lack of
sampling of *Bppov* [Lines 364-371].

*Minor criticisms:*

*- References 1 and 5 are duplicates*

**Our answer:** Thank you, we deleted the duplicated reference.

*- The Introduction starts by stating “Bordetellae are beta-proteobacteria that can be found in*
*the environment”, but then most of the described species are human- or animal-restricted,*
*which I found confusing*

**Our answer:** We modified the sentence accordingly [First line of Introduction, **Lines**
**67-68**]: “*Bordetellae* are beta-proteobacteria that are mainly associated with infection in
animals and humans, and sometimes retrieved from environmental samples.”

- *Unclear what is meant by “suboptimal services to the user’s community”*

**Our answer:** Thank you, the sentence was clarified: “This duality has led to
nomenclatural confusion and complexity for users, who may need to consult two distinct
databases” [Lines 119-120].

- *Italicisation needed of “fhaB3” (Discussion) and “fim2” (Abstract)*

**Our answer:** We modified and have harmonized throughout.

- *Do the authors use “isolate” and “strain” interchangeably? They should clarify if this is the*
*case, or select one over the other*

**Our answer:** We modified the manuscript, thank you; we use “isolates” when
referring to cultures/genomes from specific date and place of isolation, and ‘strain’ when
referring to specific genotypic or phenotypic characteristics (as is the classical usage).

Reviewers' Comments:

Reviewer #1:

Remarks to the Author:

No further comments.

Reviewer #2:

Remarks to the Author:

I thank the authors for their thorough response to my comments, and the changes they have made to the manuscript. I have only relatively minor comments on this revised version:

(A) My comments on the new sections on population structure:

The authors have added the statement:

"Genomic species were defined as groups of isolates that can be differentiated from other such groups using the complete genomic sequence-based ANI cutoff value of at 95%, which corresponds to a classically used threshold value for species delineation."

(1) The authors should specify how they calculate ANI – despite this being the basis of much of their analysis of population structure, the method used is not described. If the calculation only uses a small, conservatively defined fraction of the core genome, then the actual divergence between genomes will be underestimated.

(2) The authors should add a citation for the ANI threshold value in the statement above.

(3) The new text refers to an "ndrA" gene on L317, which should be "nrdA"

(B) My comments on the new sections on data analysis:

(1) Given the preserved order the cgMLST genes, the recombination detection approach is reasonable as an approximation. The authors should make it clear in the methods that the genes are ordered in the alignment – it should also be specified which genome is used to determine this gene order – and that this approach will miss recombinations affecting intergenic regions or accessory genes, which could be identified from a whole genome alignment.

(2) The added minimum spanning tree is a nice representation of the data. However, the key is illegible and needs replacing. It is also hard to see the ovine parapertussis genome, which should be circled.

(C) My comments on the new sections on epidemiological interpretation:

(1) "See next paragraph" on L241 actually refers to the next section – it would be helpful to explain the sampling thoroughly at this first mention, to add the reader's interpretation of all the analysis.

(2) In the response, the authors state "Horizontal gene transfer is considered rare or even absent within *B. pertussis* or between *B. pertussis* and other taxa." In the manuscript's Introduction, the authors state, "Horizontal gene transfer (HGT) is likely to occur between *Bordetella* species and lineages". Given the journal's broad readership, it would be helpful if the authors could clarify what level of recombination is expected across the species studied in the dataset, and whether their data are consistent with this expectation.

(3) Across the manuscript and supplementary material, the authors refer to "BIGSdb project id 25", "project name ...", "project n°24" and "project id 27". The reader should not be expected to remember what each of these are – they should be defined each time they are mentioned, and the references to the project identifiers made consistent throughout the text.

**REVIEWERS' COMMENTS**

**Reviewer #1** (Remarks to the Author):

No further comments.

**Our answer:** thank you

**Reviewer #2** (Remarks to the Author):

I thank the authors for their thorough response to my comments, and the changes they have
made to the manuscript. I have only relatively minor comments on this revised version:

**(A) My comments on the new sections on population structure:**

The authors have added the statement: "Genomic species were defined as groups of isolates
that can be differentiated from other such groups using the complete genomic sequence-
based ANI cutoff value of at 95%, which corresponds to a classically used threshold value for
species delineation."

(1) The authors should specify how they calculate ANI – despite this being the basis of much
of their analysis of population structure, the method used is not described. If the calculation
only uses a small, conservatively defined fraction of the core genome, then the actual
divergence between genomes will be underestimated.

**Our answer:** ANI values were computed with fastANI version 1.33, which takes into account
the entire genome assemblies. Details have been added in Methods Lines 735-736.

(2) The authors should add a citation for the ANI threshold value in the statement above.

**Our answer:** We added a link to the reference "Yoon, S.-H et al. 2017" and added a new
reference "Jain, C., Rodriguez-R, L.M., Phillippy, A.M. et al. High throughput ANI analysis of
90K prokaryotic genomes reveals clear species boundaries. *Nat Commun* 9, 5114 (2018).
<https://doi.org/10.1038/s41467-018-07641-9> » [Line 735].

(3) The new text refers to a ndrA gene on L317, which should be "nrdA"

**Our answer:** Thanks for pointing out this typo, "ndrA" was found Line 426 and was changed
to "nrdA".

**(B) My comments on the new sections on data analysis:**

(1) Given the preserved order the cgMLST genes, the recombination detection approach is
reasonable as an approximation. (A)The authors should make it clear in the methods that
the genes are ordered in the alignment – it should also be specified which genome is used to
determine this gene order – and (B) that this approach will miss recombinations affecting
intergenic regions or accessory genes, which could be identified from a whole genome
alignment.

**Our answer:**

(A). Methods already describe what is suggested here. Quote [§Methodes; §§ Development
of pan-genus *Bordetella* cgMLST scheme; Lines 586-589]: "All loci entered into the database

were given a BORD number of the format BORD000000, where the last six digits corresponded
to the BB0000 numbers used in the annotation of *B. bronchiseptica* RB50⁵⁹. Consequently,
loci in this scheme are in the same order as they appear in the reference genome of *B.*
*bronchiseptica* RB50.”

We added a sentence regarding alignment concatenation: “When running Gubbins with
cgMLST_genus data, alignments were concatenated following the loci order of the reference
genome RB50.” [Lines 718-720]

(B) In addition, we added this limitation in the discussion paragraph: “nor does it allow
accurate recombination detection, particularly in intergenic regions and accessory genes,
which are not included in the cgMLST schemes” [Lines 526-528].

(2) The added minimum spanning tree is a nice representation of the data. However, the key
is illegible and needs replacing. It is also hard to see the ovine parapertussis genome, which
should be circled.

**Our answer:** Thank you for the comment. *B. parapertussis* ovine genome is now circled in
light purple. The key was updated accordingly.

**(C) My comments on the new sections on epidemiological interpretation:**

(1) “See next paragraph” on L241 actually refers to the next section – it would be helpful to
explain the sampling thoroughly at this first mention, to add the reader’s interpretation of all
the analysis.

**Our answer:** We updated the sentence with more details: “A phylogeny of 124 *Bp* isolates
(selected to be representative of main *ptxP* branches and with a focus on macrolide resistance,
see Method section, Phylogenetic analysis) was built using the 2,038 loci of cgMLST_pertussis
scheme (**Figure 3**).” [Line 342-344]

(2) In the response, the authors state “Horizontal gene transfer is considered rare or even
absent within *B. pertussis* or between *B. pertussis* and other taxa.” **In the manuscript’s**
**Introduction**, the authors state, “Horizontal gene transfer (HGT) is likely to occur between
*Bordetella* species and lineages”. Given the journal’s broad readership, it would be helpful if
the authors could clarify what level of recombination is expected across the species studied
in the dataset, and whether their data are consistent with this expectation.

**Our answer:** We added “whereas in contrast, gene gain is rare or absent in *B. pertussis*, which
evolved mainly through gene loss²²” [Lines 176-177] to clarify this point.

As indicated in the method section, our genomic analysis is done on recombination-purged
concatenated multiple sequence alignment of core genes with both cgMLST_genus and
cgMLST_pertussis schemes. An analysis of recombination events per se was not an objective
of our study and would require a dedicated analysis, which we consider out of scope of this
work.

(3) Across the manuscript and supplementary material, the authors refer to “BIGSdb project
id 25”, “project name ...”, “project n°24” and “project id 27”. The reader should not be

expected to remember what each of these are – they should be defined each time they are
mentioned, and the references to the project identifiers made consistent throughout the text.
**Our answer:** Thank you. We added BIGSdb projects id and their given names, each time they
were mentioned in the text and in the Data availability section.